# A cascading policy learning framework for enhancing power grid resilience

Assem Sohaib Bensalah[1], Toufik Marir[1], Reda Yaich[2] and Mohamed Sedik Chebout[1]

[1] Research Laboratory on Computer Science's Complex Systems (ReLa(CS)2), University of Oum El Bouaghi, Oum El Bouaghi, Algeria
[2] IRT SystemX, Palaiseau, France



## ABSTRACT

Power grids, as critical cyber-physical systems, face increasing threats from adversarial attacks that can compromise their operational integrity. This article introduces a novel cascading policy learning framework that leverages a sequential application of three deep reinforcement learning algorithms to bolster the power grid's resilience. The framework implements a three-stage cascading approach: initially employing Proximal Policy Optimization to establish stable policy foundations, subsequently applying Trust Region Policy Optimization to ensure mathematically rigorous policy updates while maintaining performance bounds, and finally utilizing Advantage Actor-Critic to minimize policy gradient variance and optimize convergence. This sequential integration creates a robust control policy that progressively refines decision-making capabilities at each stage. Experimental validation in a simulated power grid environment evidences superior performance, with the framework achieving 84% success in maintaining continuous grid functionality for 24 h under bus-tripping attacks, exceeding baseline approaches by a considerable margin. Results confirm that this multi-stage learning strategy effectively amplifies convergence speed, maximizes cumulative rewards, and strengthens power grid resilience against cyber-physical threats.

## INTRODUCTION

The evolution of power grids into critical cyber-physical systems has created infrastructure whose failure can cascade across entire societies. Yet these systems now confront an unprecedented convergence of cyber threats (*Krause et al., 2021*; *Yohanandhan et al., 2020*), physical vulnerabilities (*Islam, Baig & Zeadally, 2019*; *Paul et al., 2021*), and natural disasters (*Mohamed et al., 2019*; *Bouramdane, 2024*) that traditional management approaches cannot adequately handle. Traditional grid management approaches are inadequate for interdependent sequential decisions, especially when solving dynamic security challenges. Artificial intelligence techniques yield a promising solution to tackle these challenges. In particular, recent advances in machine learning, such as deep reinforcement learning, reshape the research landscape for power grid control.

Corresponding authors
Assem Sohaib Bensalah,
assem.bensalah@univ-oeb.dz
Toufik Marir,
marir.toufik@univ-oeb.dz

Deep reinforcement learning (DRL), a subset of reinforcement learning (RL) that employs deep neural networks, has successfully optimized fault detection, load balancing, and real-time control systems (*Belhadi et al., 2021*; *Nakabi & Toivanen, 2021*; *Al-Saadi, Al-Greer & Short, 2023*). RL techniques also have found widespread application in bolstering the resilience of power grid systems (*Ernst, Glavic & Wehenkel, 2004*; *Lan et al., 2020*; *Subramanian et al., 2021*; *Chauhan, Baranwal & Basumatary, 2023*), which has a tangible operational impact. These include maintaining a continuous electricity supply to vital infrastructure, mitigating the ramifications of natural disasters, and safeguarding against cyber-attacks (*Gautam, 2023*). However, current approaches suffer from several limitations, including the high-dimensional nature of decision-making processes and the inability of traditional methods to adapt to dynamic and adversarial environments (*Zhu et al., 2014*; *Xie et al., 2021*).

To investigate DRL's practical applications in optimizing power grid management under targeted attacks, we utilize the Grid2Op platform as a powerful testbed (*Donnot, 2020*). Within this context, adversarial attacks in cyber-physical systems refer to deliberate actions by malicious actors to disrupt normal system operations through manipulation of control signals, data corruption, or direct physical compromise. In power grids, such attacks typically manifest as line-tripping events, load manipulation, or false data injection that can jeopardize system stability and trigger cascading failures and widespread blackouts.

Understanding the grid's capacity to handle such threats requires distinguishing between two fundamental concepts. Grid robustness represents the power grid's inherent ability to maintain stable operation and resist damage under normal variations, expected disturbances, and bounded perturbations without significant performance degradation or requiring adaptation. It focuses on static tolerance thresholds and the capacity to absorb disruptions while keeping key operational parameters within safe limits. Building upon this foundation, grid resilience encompasses the comprehensive ability of the power grid to withstand, adapt to, and rapidly recover from major adverse events, including evolving targeted attacks and unexpected disruptions, while maintaining critical functions. Resilience incorporates dynamic response capabilities, adaptive recovery mechanisms, and sequential corrective actions that prevent cascading failures and ensure prompt restoration to stable operational states, thereby minimizing the duration and impact of service interruptions. The key distinction lies in the fact that robustness emphasizes static resistance and tolerance without adaptation. While resilience encompasses robustness, it also extends to include dynamic adaptation, recovery processes, and the ability to learn from and respond to evolving threats through active reconfiguration and corrective measures. This conceptual framework is particularly relevant in our DRL approach, where agents must not only withstand initial attacks (robustness) but also demonstrate resilience by taking sequential adaptive actions to reconfigure the grid and prevent system-wide failures.

In this work, we propose a novel framework to fortify the resilience of power grid systems. This framework utilizes a cascading policy that combines several deep reinforcement learning techniques, integrating Advantage Actor-Critic (A2C), Trust

Region Policy Optimization (TRPO), and Proximal Policy Optimization (PPO). As a result, this approach improves the resilience of power grid systems through three key contributions:

1. **Novel Cascading Policy Learning Framework (Methodology):** We introduce a three-stage sequential training methodology that synthesizes A2C, TRPO, and PPO techniques through a theoretical knowledge transfer mechanism, enabling progressive policy improvement and adaptive decision-making in dynamic grid scenarios under adversarial conditions.

2. **Comprehensive Empirical Validation (Experimental Results and Analysis):** We provide systematic experimental evaluation demonstrating superior performance over single-algorithm and dual-stage baseline approaches, including ablation studies across all six algorithmic permutations and statistical significance testing using 100 adversarial scenarios on the Grid2Op platform.

3. **Open-Source Implementation and Full Reproducibility (Code Availability):** We deliver a complete, permanent, and version-pinned open-source release including source code, environment configurations, hyperparameter settings, trained model weights, and evaluation scripts to ensure full reproducibility and foster future research by the community.

The remainder of this article is as follows: 'Related Works' presents related works that focus on using RL techniques to improve the resilience of power grids. Subsequently, 'Methodology' introduces the proposed framework. 'Experimental Results and Analysis' then presents the results, including a description of the simulated environment, the experimental setup, implementation details, and performance analysis. 'Discussion' discusses the results, and 'Conclusion' concludes the article.

## RELATED WORKS

The evolution of power grid management has progressed through distinct methodological paradigms, each contributing valuable insights while revealing specific constraints that motivate our framework.

Historically, conventional power grid control has relied primarily on model-based approaches such as Optimal Power Flow (OPF) and Model Predictive Control (MPC) (*Faulwasser et al., 2018*; *Diab, Abdelhamid & Sultan, 2024*), which have maintained dominance in industrial applications due to their mathematical rigor and proven performance under well-characterized operating conditions. These approaches excel when system dynamics are well understood and operating conditions remain within the predicted parameters, providing deterministic solutions with established theoretical guarantees. However, these model-based approaches encounter elemental restrictions when confronting modern grid challenges, particularly in adversarial scenarios. Their reliance on accurate system models becomes a critical vulnerability when encountering unforeseen, fast-acting attacks that exploit non-linear system behaviors or manipulate the very sensors upon which these models depend. The computational requirements for

solving non-linear optimization problems in real-time further constrain their applicability in dynamic threat environments where rapid response is essential. Most critically, classical approaches struggle with the speed and uncertainty characteristic of adversarial scenarios, as they assume system behavior follows predictable patterns that targeted attacks deliberately violate.

Recognizing the drawbacks of model-based methods, researchers started exploring reinforcement learning as an alternative paradigm for learning control policies directly from system interactions without requiring explicit models. *Ernst, Glavic & Wehenkel (2004)* pioneered the use of RL in power systems, investigating its capability as a stability control framework and establishing RL's potential for handling uncertainties in dynamic power system environments. This foundational research revealed that agents could adapt to changing conditions through experience, laying the groundwork for subsequent developments in the field. Building upon this foundation, *Lan et al. (2020)* advanced the methodology by introducing a dueling deep Q-network (Dueling DQN) to maximize available transfer capacity through optimal topology control. Their approach incorporated imitation learning for initial policy generation and guided exploration for training, showing improved performance over conventional methods in controlled scenarios. Despite these promising results, early RL approaches remained constrained by severe bottlenecks. *Ernst, Glavic & Wehenkel*'s *(2004)* work was limited to offline learning scenarios, which proved impractical for real-time grid management. While *Lan et al.*'s *(2020)* methodology required extensive pre-training data, it caused consequential impediments in dynamic environments where historical patterns may not reflect current conditions. *Subramanian et al. (2021)* further investigated simplified DRL approaches using cross-entropy methods for power flow control, establishing baseline performance metrics while revealing the inherent trade-offs between computational simplicity and scalability to large-scale grid systems. Yet, translating this to practice was difficult due to poor training efficiency, high sample complexity, and a lack of stability guarantees.

As the field matured, researchers developed more engineered DRL architectures to address the drawbacks of early approaches. The Learning to Run a Power Network (L2RPN) challenge, introduced by *Marot et al. (2020)*, established standardized frameworks for evaluating RL approaches in power grid topology control, focusing on grid capacity optimization through bus reconfigurations. This initiative catalyzed noteworthy research progress and fostered systematic comparison of different methodologies. Subsequent work by *Lehna et al. (2023)* compared rule-based agents with PPO-based approaches (*Schulman et al., 2017*), revealing important behavioral patterns across different operational scenarios. *Chauhan, Baranwal & Basumatary (2023)* built upon these findings through PowRL, combining heuristic approaches with RL for topology optimization while maintaining robust operation under uncertain conditions. *Dorfer et al. (2022)* proposed AlphaZero-based agents (*Silver et al., 2018*) for topology reconfiguration, illustrating cost-effective alternatives to classical congestion management. These developments showed that state-of-the-art DRL architectures could

achieve superior performance in grid optimization tasks, though they typically relied on single-algorithm paradigms that inherited the inherent constraints of their chosen approach.

Addressing the growing threat landscape, recent research has explored adversarial learning frameworks specifically designed for the security of the power grid. *Chen, Nguyen & Hassanaly (2024)* developed a novel adversarial multi-agent reinforcement learning (MARL) framework for detecting evolving false data injection attacks, employing competing attacker and defender agents in continuous learning environments. Their defender agent, implemented as a deep Q-network, achieved 98.7% detection accuracy for previously unseen FDIAs while maintaining sub-200 ms detection latency, validating the potential of adversarial training for cybersecurity applications. Similarly, *Mukherjee et al. (2023)* introduced federated reinforcement learning methodologies for amplifying cyber resiliency in networked microgrids, developing multi-agent federated Soft Actor-Critic algorithms that address data-sharing concerns while optimizing microgrid interconnectedness. These approaches represented tangible advances in applying RL to adversarial scenarios, showing that agents could learn to detect and respond to specific attack patterns through exposure during training. However, while these adversarial frameworks excel at detecting and countering the specific threat models their agents learned to counter, they are limited by critical bottlenecks in generalizability. Their effectiveness remains tied to the predefined adversarial scenarios encountered during training, potentially leaving systems vulnerable to novel attack methodologies that differ from training distributions. This specificity-generalization trade-off represents a persistent challenge in adversarial learning for critical infrastructure such as power grids.

Analysis of existing methodologies reveals that individual RL algorithms, while evidencing promise, suffer from specific constraints when applied in isolation to power grid control. PPO-based approaches often experience initial instability during the crucial early learning phases, potentially compromising system safety during policy development. TRPO provides theoretical guarantees for monotonic improvement but incurs substantial computational overhead that hampers real-time deployment in large-scale systems. A2C offers computational efficiency but lacks the stability guarantees necessary for critical infrastructure applications. Existing approaches typically commit to a single algorithm, inheriting its specific weaknesses without mechanisms to mitigate them. Furthermore, current frameworks lack effective transfer learning mechanisms between different learning stages or algorithmic components. Each algorithm trains from scratch or with limited initialization, resulting in inefficient policy development and prolonged learning curves. This inefficiency is particularly problematic in power grid applications where extensive training on live systems is impractical and simulation-to-reality gaps can compromise performance. The adversarial learning approaches discussed above, while advancing the state-of-the-art in attack detection, exhibit a key shortcoming in their focus on specific threat models. Game-theoretic methods, though theoretically sound, suffer from computational complexity that prevents real-time deployment in

**Table 1 Prior research on DRL applications in resilient power grid management.**

| Study | Main contribution | Limitations |
|---|---|---|
| *Faulwasser et al. (2018)*, *Diab, Abdelhamid & Sultan (2024)* | Model-based control (OPF, MPC) for grid optimization | Model-dependent; computationally expensive; vulnerable to adversarial attacks |
| *Ernst, Glavic & Wehenkel (2004)* | First RL framework for power stability | Limited to offline learning scenarios |
| *Lan et al. (2020)* | Dueling DQN with imitation learning | Requires extensive pre-training data |
| *Subramanian et al. (2021)* | Cross-entropy method for topology control | Constrained action space |
| *Chauhan, Baranwal & Basumatary (2023)* | PowRL with heuristic-guided learning | Arduous implementation requirements |
| *Dorfer et al. (2022)* | AlphaZero for congestion management | High computational overhead |
| *Lehna et al. (2023)* | Rule-based and PPO-based approaches | Limited novel methodological advances |
| *Marot et al. (2021)* | DRL framework for N−1 contingency | Limited grid generalization |
| *Chen, Nguyen & Hassanaly (2024)* | Adversarial MARL for FDIA detection | Cannot guarantee detection of all FDIA variants |
| *Mukherjee et al. (2023)* | Federated RL for microgrid cyber resilience | Training instability and limited attack scenarios |

large-scale systems. More critically, approaches optimized for detecting particular attack types struggle to generalize to diverse or evolving threat landscapes without extensive retraining.

Table 1 provides a comprehensive comparison of existing approaches in DRL-based power grid management, highlighting their key contributions and inherent limitations that motivate our cascading framework design.

This comprehensive analysis reveals three critical research gaps that existing methodologies fail to address systematically:

1. **Limited integration of multiple learning paradigms:** Current approaches typically rely on single RL algorithms, failing to leverage the complementary strengths of different learning methodologies or provide mechanisms to compensate for individual algorithmic weaknesses.

2. **Lack of systematic knowledge transfer:** Existing frameworks lack effective transfer learning between different algorithmic stages or training phases, resulting in inefficient policy development, prolonged training times, and suboptimal performance across diverse operating conditions.

3. **Insufficient generalization for adversarial scenarios:** While recent work addresses specific attack types through adversarial training or detection mechanisms, no existing framework provides comprehensive operational resilience against diverse, evolving threat landscapes without requiring extensive retraining or attack-specific adaptations. Current approaches focus on identifying and countering specific threats rather than maintaining stable operations regardless of the origin of disruption.

These gaps collectively mandate the development of a unified framework that systematically integrates multiple learning paradigms, supports efficient knowledge transfer across training stages, and develops general control policies that maintain grid

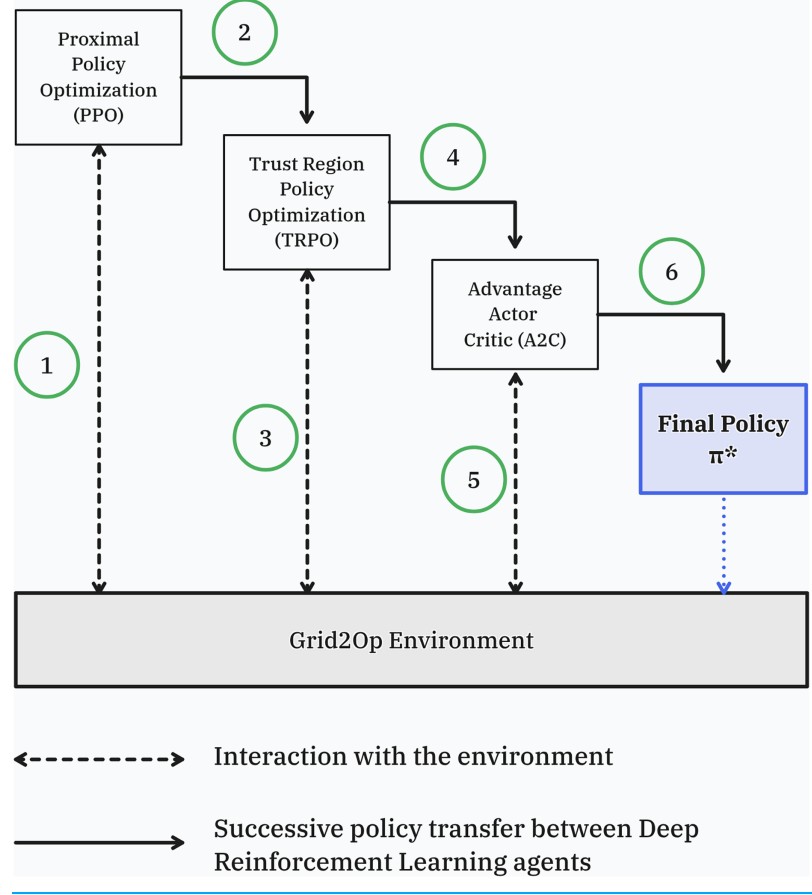

**Figure 1** **Architecture of the cascading policy learning framework (CPLF).** The framework sequentially transfers learned policy parameters from proximal policy optimization (PPO) (1) to trust region policy optimization (TRPO) (3) and subsequently to advantage actor-critic (A2C) (5), as shown by the transfer arrows (2, 4). Each agent interacts with the environment to progressively optimize the policy, culminating in a final, optimized policy $\pi^*$ (6).

stability under diverse adversarial conditions without requiring attack-specific customization.

## METHODOLOGY

The Cascading Policy Learning Framework (CPLF), depicted in Fig. 1, introduces a novel sequential training approach that leverages the complementary strengths of multiple DRL algorithms within a simulated power grid environment. This framework systematically unifies three state-of-the-art DRL algorithms to develop a robust power grid control policy: Proximal Policy Optimization (*Schulman et al., 2017*), Trust Region Policy Optimization (*Schulman, 2015*), and Advantage Actor-Critic (*Mnih, 2016*).

### Theoretical foundation of cascading knowledge transfer

The CPLF consists of three main steps that implement progressive knowledge distillation, where each algorithm acts as a teacher for the subsequent one, following the established teacher-student paradigm (*Hinton, 2015*). This cascading approach goes beyond sequential

execution by embedding knowledge transfer as a core design feature. From a transfer learning perspective, the framework cultivates adaptive policies for increasingly dynamic grid scenarios, with the knowledge accumulated in earlier stages informing the learning process of subsequent algorithms.

The mathematical foundation of this cascading approach lends itself to formalization through the lens of transfer learning. Let $\theta_1$, $\theta_2$, and $\theta_3$ represent the parameter sets for PPO, TRPO, and A2C, respectively. The transfer process follows: $\theta_2^{(0)} = \theta_1^{(f)}$ and $\theta_3^{(0)} = \theta_2^{(f)}$ where the superscripts (0) and (f) denote initial and final parameter states. This sequential inheritance allows each stage to leverage the knowledge gained from previous stages, while also applying its distinct optimization constraints. At each stage transition, we transfer the complete actor and critic network parameters (weights and biases) but reinitialize the optimizer state. This design choice allows each algorithm's distinct optimization dynamics to take full effect while preserving the learned control strategy.

Hence, the three sequential stages are:

1. **Initial policy development:** Proximal Policy Optimization establishes stable baseline behavior through its clipped surrogate objective function:

$$L^{\text{CLIP}}(\theta) = \mathbb{E}_t\big[\min\big(r_t(\theta)\hat{A}_t, \text{clip}(r_t(\theta), 1 - \varepsilon, 1 + \varepsilon)\hat{A}_t\big)\big]$$

   with $\varepsilon = 0.2$ constraining policy updates. This stability is pivotal for creating a robust initial policy within the highly interconnected power grid environment. The initial PPO stage then acts as a foundational teacher, establishing stable behavioral patterns.

2. **Policy fine-tuning:** Trust Region Policy Optimization implements mathematically rigorous updates by enforcing Kullback-Leibler (KL) divergence constraints, a statistical measure that quantifies the difference between two probability distributions, building upon PPO's trained policy network weights:

$$\max_\theta \mathbb{E}_t\left[\frac{\pi_\theta(a_t \mid s_t)}{\pi_{\theta_{\text{old}}}(a_t \mid s_t)}\hat{A}_t\right] \quad \text{subject to} \quad D_{\text{KL}}(\pi_{\theta_{\text{old}}}, \pi_\theta) \leq \delta$$

   with $\delta = 0.01$ sustaining policy stability. This constraint guarantees that the policy variance reduction preserves the useful policy patterns learned in the initial stage while optimizing for unstable grid scenarios, particularly those involving multiple concurrent contingencies or rapid load fluctuations. TRPO then hones these patterns, leveraging its trust region approach to deliver robust policy improvements.

3. **Final optimization:** Advantage Actor-Critic further optimizes the policy using a learned value function baseline, initialized with TRPO's trained policy network weights:

$$L^{\text{A2C}}(\theta) = \mathbb{E}_t[\log \pi_\theta(a_t \mid s_t)(R_t - V(s_t)) + \alpha H(\pi_\theta)]$$

   with $V(s_t)$ normalizing returns across states. This approach balances exploration and exploitation, normalizing returns across different grid states and resulting in more precise policy improvements, primarily in scenarios with diverse load patterns and grid topologies.

---

**Algorithm 1**  Cascading policy learning framework.

**Input:** Environment $\mathscr{E}$, PPO iterations $N_{PPO}$, TRPO iterations $N_{TRPO}$, A2C iterations $N_{A2C}$
**Initialize:** PPO, TRPO, A2C parameters
**Output:** Optimized policy $\pi_\theta^*$
// Single Algorithm Training Phase
Initialize $\pi_\theta^{\mathrm{PPO}}$ randomly
**for** iteration = 1 to $N_{PPO}$ **do**
    Collect trajectories using current policy $\pi_\theta^{\mathrm{PPO}}$
    Update policy using PPO objectives
**end for**
// Policy Transfer and TRPO Training Phase
Initialize $\pi_\theta^{\mathrm{TRPO}} \leftarrow \pi_\theta^{\mathrm{PPO}}$ {Transfer learned policy}
**for** iteration = 1 to $N_{TRPO}$ **do**
    Collect trajectories using current policy $\pi_\theta^{\mathrm{TRPO}}$
    Update policy using TRPO objectives with conservative updates
**end for**
// Policy Transfer and A2C Fine-tuning Phase
Initialize $\pi_\theta^{\mathrm{A2C}} \leftarrow \pi_\theta^{\mathrm{TRPO}}$ {Transfer learned policy}
**for** iteration = 1 to $N_{A2C}$ **do**
    Collect trajectories using current policy $\pi_\theta^{\mathrm{A2C}}$
    Update policy using A2C objectives
**end for**
**return** $\pi_\theta^{\mathrm{A2C}}$ as $\pi_\theta^*$

---

The knowledge transfer between algorithmic stages represents a critical component of the CPLF framework. Each transition involves the inheritance of the complete policy network parameters, where the successor algorithm initializes its neural network weights and biases directly from the trained predecessor model. Specifically, this transfer is intentionally limited to the policy and value function weights and does not include the optimizer's state (*e.g.*, Adam's moments). Consequently, each algorithmic stage begins with a re-initialized optimizer. This approach solidifies maximal knowledge retention of the learned control strategy while allowing the distinct optimization dynamics of each algorithm to take full effect without being constrained by the momentum of a previous stage. The transfer process leverages the identical network architectures maintained across all stages, allowing for seamless parameter mapping without dimensional conflicts.

The constraints of each algorithm implicitly regularize the transfer process. PPO's clipping mechanism ($\varepsilon = 0.2$) maintains initial stability. TRPO's KL divergence constraint ($\lambda = 0.01$) maintains learned behaviors through mathematically rigorous updates, and A2C's value function baseline empowers precise policy improvements by reducing gradient variance.

Formally, the policy transfer mechanism minimizes the divergence between the source policy $\pi_{\mathrm{source}}$ and target policy $\pi_{\mathrm{target}}$: $\min_\theta D_{\mathrm{KL}}(\pi_{\mathrm{source}} \| \pi_{\mathrm{target}})$, subject to performance constraints that guarantee the target policy maintains or improves upon the performance of the source policy. This theoretical foundation guarantees that our cascading approach optimizes both policy performance and knowledge retention across algorithmic transitions.

The numbered arrows in Fig. 1 illustrate the sequence of operations. Hence, arrows (1, 3, and 5) represent the interaction of each algorithm with the environment. Arrows 2

and 4 represent the transfer of policy network weights between successive algorithms. Finally, arrow 6 depicts the final policy deployment by producing the optimal policy $\pi$.

Algorithm 1 details the proposed framework.

### Network architecture and data handling

Our consistent neural network architecture allows for seamless parameter transfer across all algorithmic stages. The network features two fully connected hidden layers with 1,200 and 1,000 units, respectively, utilizing LeakyReLU activation functions with $\alpha = 0.01$. This architectural choice delivers sufficient representational capacity for the high-dimensional state space while minimizing resource consumption. Complete neural network parameter transfer occurs between stages, including weights and biases from the predecessor algorithm, with no layer freezing, allowing complete adaptation while preserving learned representations.

Hyperparameter selection followed a systematic manual tuning process informed by values commonly reported in DRL literature. Starting from the defaults of stable-baselines3 (*Raffin et al., 2021*), we iteratively adjusted the learning rates, batch sizes, and iteration counts based on the agent's performance on validation scenarios. While this approach yielded stable, high-performing policies, we acknowledge that automated hyperparameter optimization (*e.g.*, Bayesian optimization *via* Optuna (*Akiba et al., 2019*)) could further improve performance and represents a valuable direction for future work.

The data for our experiments comes from the Grid2Op environment in a structured format that minimizes preprocessing requirements while maintaining the ontological integrity of power system measurements (*Donnot, 2020*). Data points are directly integrated into the reinforcement learning agents without requiring transformational procedures such as normalization or feature scaling, thus preserving their ontological integrity. This methodological approach maintains a streamlined operational footprint while mitigating the risk of systematic biases that might otherwise arise through data manipulation techniques.

## EXPERIMENTAL RESULTS AND ANALYSIS

To evaluate the proficiency and robustness of CPLF, we trained the agent within a Grid2Op Robustness 2020 challenge environment. The agent was trained on a mid-level difficulty setting and then tested on the most difficult "competition level" to assess its ability to generalize and maintain grid stability against destabilizing attacks.

### Experimental setup

Grid2Op (*Donnot, 2020*) serves as the foundational framework for our resilient grid control experiments. It models the power grid control dynamics as a Markov Decision Process (MDP), which is well-suited for RL. Its compatibility with the OpenAI Gym library (*Brockman et al., 2016*) further optimizes its flexibility, supporting the development and assessment of a diverse range of control agents, including RL models, heuristics, and optimization algorithms. We employed the Grid2Op Robustness 2020 challenge environment (*Marot et al., 2020*), which implements a modified subset of the

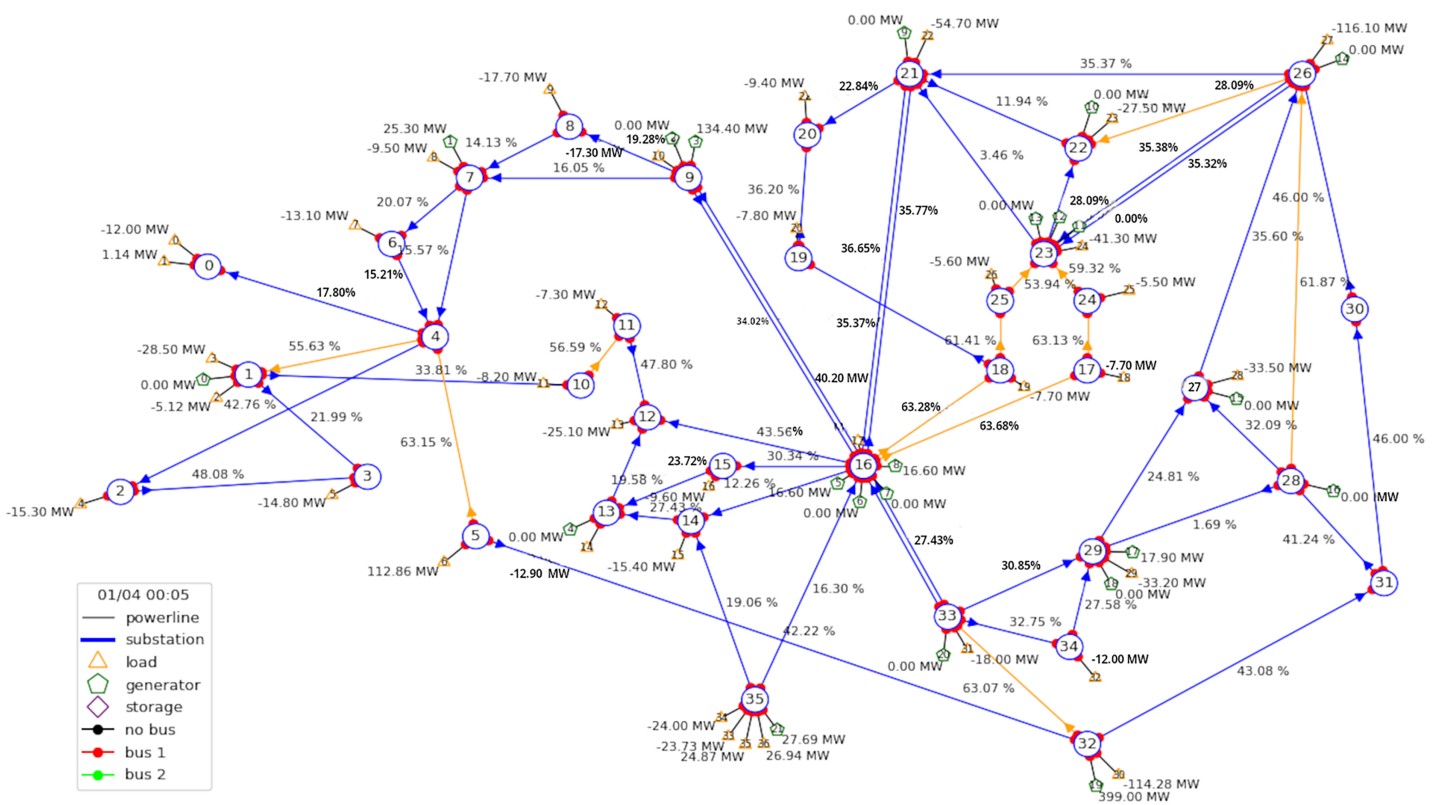

**Figure 2 A schematic shows the Grid2Op robustness 2020 challenge testbed, which modifies an IEEE 118-bus system.** This environment simulates dynamic real-world and threat scenarios. It comprises 36 substations, 59 power lines, 22 generators, and 37 loads.

**Table 2 Grid2Op environment configuration.**

| Component | Specification | Details |
|---|---|---|
| Python version | 3.11.13 | Programming language environment |
| Grid2Op version | 1.12.1 | Pinned for reproducibility |
| Backend (LightSim2Grid) | 0.10.3 | Fast power flow computation |
| ML framework | Stable-Baselines3 (2.1.0) | Reinforcement learning library |
| Deep learning backend | PyTorch (2.6.0) + CUDA (12.4) | For GPU-accelerated training |
| Base system | a modified IEEE-118 | 36 substations, 59 lines |
| Observation space | 765 dimensions | Table 3 breakdown |
| Action space | 66,000+ discrete actions | Set line operations |
| Reward function | Custom survival-based | 'CPLF's reward function' definition |
| Attack model | Line-tripping attacks | $\rho$-normalized, 4 h duration |

IEEE 118-bus system and uses only traditional energy sources, as depicted in Fig. 2. This environment comprises a total of 36 substations managing power distribution, 59 powerlines for power transmission, 22 generators supplying power, and 37 loads, including interconnections with other grid sections represented as negative loads. Table 2 provides

**Table 3 Observations for the cascading policy learning framework.**

| Category | Attribute | Description |
|---|---|---|
| Temporal attributes | month | The present month. |
| | day_of_week | The present day of the week (Monday = 0, Sunday = 6). |
| | day | The present day of the month (1 = first day). |
| | hour_of_day | The present day's hour (0–23). |
| | minute_of_hour | The present minute within the current hour (0–59). |
| Generator attributes | gen_p | Active power output of each generator (MW). |
| | gen_q | Reactive power output of each generator (MVar). |
| | gen_v | Voltage level at the bus linked to each generator (kV). |
| Load attributes | load_p | Active power consumption of each load (MW). |
| | load_q | Reactive power consumption of each load (MVar). |
| | load_v | Voltage level at the bus linked to each load (kV). |
| Powerline origin attributes | p_or | Active power flow at the origin end of each powerline (MW). |
| | q_or | Reactive power flow at the origin end of each powerline (MVar). |
| | v_or | Voltage level at the origin end of each powerline (kV). |
| | a_or | Current flow at the origin end of each powerline (A). |
| Powerline extremity attributes | p_ex | Active power flow at the extremity end of each powerline (MW). |
| | q_ex | Reactive power flow at the extremity end of each powerline (MVar). |
| | v_ex | Voltage level at the extremity end of each powerline (kV). |
| | a_ex | Current flow at the extremity end of each powerline (A). |
| Bus info | rho | Utilization rate of each powerline (current flow relative to thermal limit). |
| | line_status | Operational status (connected or disconnected) of each powerline. |
| | timestep_overflow | The number of timesteps a powerline has overloaded. |
| Topology info | topo_vect | Bus connections for each element (load, generator, powerline ends). |
| Cooldowns | time_before_cooldown_line | Remaining timesteps before a powerline can be interacted with again. |
| | time_before_cooldown_sub | Remaining timesteps before a substation can be interacted with again. |
| Maintenance | time_next_maintenance | Timesteps until the next scheduled maintenance for each powerline. |
| | duration_next_maintenance | Duration of the upcoming maintenance for each powerline. |

the complete technical specifications of the experimental environment, including the Python version, Grid2Op version, backend details, observation space dimensions, action space size, and attack model parameters. It is worth noting that the detailed attributes of the observation space are presented in Table 3.

Furthermore, this testbed supports a high-dimensional action space comprising over 66,000 discrete and continuous actions, allowing for end-to-end exploration and optimization of power grid performance. It delivers rich observational data to agents, including temporal information such as the month, day, and hour; power flow measurements, *e.g.*, active/reactive power, voltage, and current, at both ends of the power lines; generator and load statistics, including active power output and voltage levels; grid topology status, for instance, bus connections, line status, and capacity; operational constraints, namely cooldown timers for powerlines and substations; and finally maintenance schedules for powerlines.

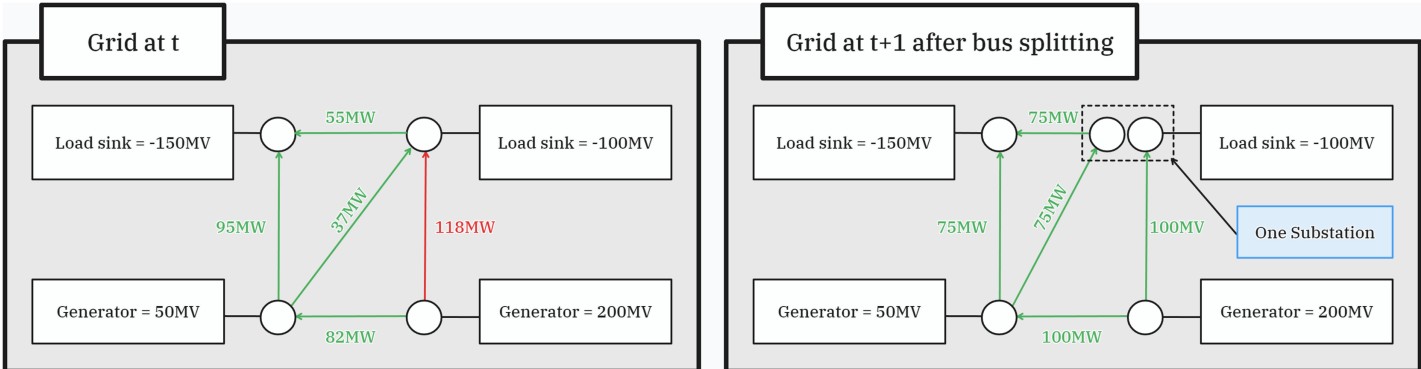

**Figure 3** **Power flow redistribution through bus-splitting: left panel (grid at time *t*): initial grid state showing line congestion of 118 MW (red line) exceeding operational limits due to concentrated power flows from 250 MW total generation to −250 MW aggregate load demand.** Right panel (grid at time *t* + 1): post-intervention state after bus-splitting at the receiving substation creates parallel power flow paths, reducing maximum line loading to 100 MW while maintaining identical generation dispatch and load consumption. 

Agents can interact with this environment through two primary action categories: "Change" operations, which toggle powerline connections and bus assignments, and "Set" operations, which directly assign specific statuses or buses to grid components. The powerline status (connected/disconnected) can be modified either by directly setting its status using "set_line_status" or by assigning components to specific buses *via* "set_bus", which indirectly affects line connectivity based on topological configurations. To maintain consistent action semantics and simplify the learning process, we customized the agent's action space to use "Set" actions ("set_line_status" and "set_bus"), providing deterministic state transitions that yield more stable policy learning compared to the toggle-based "Change" operations. These actions encompass a wide range of grid management capabilities, including line status modification, topological alterations, power redispatch, load curtailment, and management of storage units. This framework also supports predefined and custom reward functions for performance evaluation, such as CloseToOverflowReward, which penalizes states where lines approach overflow conditions, and LinesCapacityReward, which evaluates the optimal use of line capacity.

Some environments include opponents that simulate disruptive attacks. In our simulated environment, the adversary quasi-randomly performs line-tripping attacks. Specifically, the adversary employs a weighted random selection mechanism where line-tripping probabilities are proportional to their strategic importance ($\rho$-normalized), creating realistic adversarial behavior that targets vulnerable infrastructure while maintaining stochasticity. The attacks occur with a 24-h cooldown period and a 4-h duration, subject to a budget constraint limiting their frequency and intensity.

For a direct comparison with trained solutions, Grid2Op supplies baseline agents. For instance, the "RandomAgent" takes random actions, while the "ExpertAgent" uses a greedy algorithm to streamline the overflow resolution process.

One critical challenge these agents must solve is preventing power grid capacity overload. It can occur due to a combination of factors, such as scheduled maintenance events, equipment faults, and malicious actions. Additionally, fluctuations in power

demand, extreme weather conditions, and inadequate infrastructure can also exacerbate such incidents. It is indispensable that when we train control agents, they must be capable of proficiently managing overflow situations using the actions available. Figure 3 illustrates a bus-splitting action in response to an overflow scenario within a simulated Grid2Op environment. As depicted on the left side of the Fig. 3, at timestep $t$, the grid experiences line congestion of 118 MW on the eastern transmission path (shown in red), exceeding operational limits due to aggregate power demands of $-150$ and $-100$ MV at the load sinks. The implemented topological action involves bus-splitting at the receiving substation, creating parallel power flow paths. This reconfiguration at $t + 1$, as visualized on the right side of Fig. 3, redistributes power flows more optimally across the network, reducing maximum line loading to 100 MW while maintaining constant generation dispatch (50, 200 MV) and load consumption patterns. The bus-splitting intervention successfully mitigated the congestion without requiring changes to generation schedules or load curtailment, which can be costly and detrimental to system reliability.

## CPLF's reward function

We define the composite reward function $R_t$ at timestep $t$ as a weighted linear combination of four reward components:

$$R_t = w_1 \cdot R_t^{N-1} + w_2 \cdot R_t^{game} + w_3 \cdot R_t^{capacity} + w_4 \cdot R_t^{alert} \tag{1}$$

where the weight vector $\mathbf{w} = [3.0, 2.0, 1.5, 1.0]$ establishes a hierarchical priority structure that emphasizes security criteria over operational efficiency.

### *Reward components*

**N−1 security criterion** ($R_t^{N-1}$, **weight = 3.0**) The N−1 reward evaluates grid resilience by simulating single-line contingencies, a main principle of security in power system operations. For each powerline $\ell$, this component computes:

$$R_t^{N-1}(\ell) = f\left(\max_{i \in \mathscr{L} \backslash \{\ell\}} \rho_i^{(\ell)}\right) \tag{2}$$

where $\rho_i^{(\ell)}$ represents the thermal loading ratio of line $i$ following the disconnection of line $\ell$, and $\mathscr{L}$ denotes the set of all powerlines. The function $f(\cdot)$ maps the maximum post-contingency flow to a reward value, with higher flows yielding lower rewards. This component receives the highest weight (3.0) as maintaining N−1 security is paramount for critical infrastructure resilience, particularly under adversarial conditions where attackers may strategically target vulnerable lines.

**Operational stability** ($R_t^{game}$, **weight = 2.0**) The gameplay reward provides immediate feedback on the agent's operational status:

$$R_t^{game} = \begin{cases} r_{failure} < 0 & \text{if blackout occurs} \\ r_{violation} = \frac{r_{failure}}{2} & \text{if operational rules violated} \\ r_{nominal} > 0 & \text{otherwise} \end{cases} \tag{3}$$

This component penalizes catastrophic failures (blackouts) and rule violations while rewarding stable operation. The intermediate penalty for rule violations (half the failure penalty) encourages the agent to avoid technical infractions without treating them as severely as complete system failures. With a weight of 2.0, this component ensures basic operational viability while remaining subordinate to security considerations.

**Line capacity management** ($R_t^{capacity}$, **weight = 1.5**) The capacity reward incentivizes efficient utilization of transmission infrastructure through a linear relationship with line loading:

$$R_t^{capacity} = r_{max} - (r_{max} - r_{min}) \cdot \frac{1}{|\mathscr{L}_{active}|} \sum_{i \in \mathscr{L}_{active}} \frac{I_i}{I_i^{max}} \tag{4}$$

where $\mathscr{L}_{active}$ represents connected powerlines, $I_i$ is the current flow on line $i$, and $I_i^{max}$ is its thermal limit. This formulation considers only active lines, preventing disconnected lines from artificially inflating the reward. The linear structure (as opposed to quadratic alternatives) provides consistent gradient signals across varying loading conditions. The moderate weight (1.5) positions capacity optimization as important but subordinate to security and stability.

**Predictive alert mechanism** ($R_t^{alert}$, **weight = 1.0**) The alert reward implements a sparse, delayed reward structure for proactive threat detection:

$$R_t^{alert} = \begin{cases} r_{max}^{blackout} = 2.0 & \text{if alert sent AND blackout within } \tau \text{ steps} \\ r_{min}^{blackout} = -10.0 & \text{if no alert AND blackout within } \tau \text{ steps} \\ r_{max}^{safe} = 1.0 & \text{if no alert AND survival beyond } \tau \text{ steps post}-\text{attack} \\ r_{min}^{safe} = -1.0 & \text{if alert sent BUT survival beyond } \tau \text{ steps post}-\text{attack} \\ r_{bonus} = 1.0 & \text{if episode completed successfully} \\ 0 & \text{otherwise} \end{cases} \tag{5}$$

where $\tau$ represents the alert time window (typically 12 steps). This component exhibits two critical characteristics:

- **Delayed reward structure:** The reward is received $\tau$ timesteps after alert submission, requiring the agent to develop temporal credit assignment capabilities.
- **Sparse reward distribution:** Non-zero rewards occur only during attacks and blackouts, creating a challenging learning signal that necessitates efficient exploration strategies.

The alert mechanism encourages predictive behavior by heavily penalizing missed threats ($r_{min}^{blackout} = -10.0$) while moderately discouraging false alarms ($r_{min}^{safe} = -1.0$). The asymmetric penalty structure reflects the critical nature of power grid operations, where failing to anticipate a blackout-inducing attack is significantly more consequential than issuing unnecessary warnings. The lowest weight (1.0) acknowledges that while predictive capabilities enhance resilience, they remain supplementary to fundamental security and operational requirements.

**Table 4 Key performance indicators (KPIs) for resilience evaluation.**

| KPI | Unit | Definition |
| --- | --- | --- |
| Survival time | Timesteps | Total number of timesteps the agent maintains grid stability without blackout (max: 2,000) |
| Success rate | Percentage (%) | Proportion of scenarios where agent survives $\geq$288 timesteps (24 h) |
| Failure rate | Percentage (%) | Proportion of scenarios where agent experiences blackout before 288 timesteps |
| Median survival | Timesteps | 50th percentile of survival duration distribution across all scenarios |
| 95th percentile survival | Timesteps | 95th percentile of survival duration, indicating high-end performance |
| Constraint violations | Count | Number of episodes where thermal overloads ($\rho > 1.0$) persist beyond allowed duration |
| Performance consistency | Ratio | Proportion of scenarios achieving "Good" or "Excellent" performance (survival >288 timesteps) |

### Design rationale

The hierarchical weighting scheme [3.0, 2.0, 1.5, 1.0] embodies a principled approach to multi-objective reinforcement learning in critical infrastructure domains:

- **Security-first paradigm:** The highest weight on N−1 security ensures agents prioritize grid resilience, particularly relevant when defending against adversarial attacks that exploit contingency vulnerabilities.

- **Stability as foundation:** The secondary weight on operational stability prevents the agent from pursuing aggressive optimization strategies that might compromise basic grid functionality.

- **Efficiency as refinement:** The tertiary weight on capacity management encourages resource optimization without allowing efficiency concerns to override security imperatives.

- **Prediction as enhancement:** The quaternary weight on alert positions proactive threat detection as a valuable but non-essential capability, preventing the sparse reward structure from dominating the learning dynamics.

This composite reward function enables the CPLF framework to learn policies that balance multiple operational objectives while maintaining the security-centric focus essential for adversarial resilience in power grid management.

### Implementation and evaluation protocol

We rigorously assess the proposed CPLF framework using a scenario-based evaluation protocol within the Grid2Op simulation environment. Our primary evaluation metric is the success rate. Table 4 formally defines all key performance indicators (KPIs) used throughout our evaluation. This metric, as specified in Eq. (6), quantifies resilience as the percentage of simulated scenarios in which agents maintain grid stability for a minimum duration of 288 timesteps, equivalent to 24 h of continuous operation. This duration captures a complete daily load cycle, creating a meaningful timeframe to assess the agent's performance across varying operational conditions, including peak and off-peak demand periods. The 24-h evaluation window allows for observation of agent behavior under typical fluctuations in electricity demand and traditional generation that occur throughout

the day, establishing it as a practical benchmark for testing grid management strategies in simulation environments.

$$SuccessRate = \frac{\text{Number of scenarios surviving} \geq 288 \text{ timesteps}}{\text{Total scenarios}} \times 100\%. \qquad (6)$$

Grid stability maintenance requires preventing both thermal overloads ($\rho > 1.0$ for extended periods) and topological instabilities that result in disconnected loads or unsustainable power flow distributions. The evaluation encompasses 100 distinct adversarial test scenarios, each executing for a maximum of 2,000 timesteps with systematically varied environmental and agent seed values to achieve methodological robustness and reproducible results.

We implemented systematic seed management across different testing phases. The environment evaluation utilized seeds 300–399, while algorithm-specific evaluations employed distinct seed ranges: the PPO phase (150–249), the TRPO phase (200–299), the A2C phase (300–399), and reference implementations (400–499). The observation space, detailed in Table 3, included power flow metrics ($P$, $Q$, $V$, $A$) for transmission line monitoring, generator output and load demand states, network topological configurations, line capacity utilization coefficients ($\rho$), and temporal indicators for maintenance scheduling and cooldown periods.

Through systematic hyperparameter optimization, we developed a neural network architecture featuring two fully connected hidden layers with 1,200 and 1,000 units, respectively, utilizing LeakyReLU activation functions ($\alpha = 0.01$). The network implemented a custom Actor-Critic Policy. We leveraged Stable Baselines 3 for implementing PPO, A2C, and TRPO algorithms (*Raffin et al., 2021*). All experiments were conducted on an NVIDIA Tesla P100 GPU platform. Table 5 outlines the complete set of hyperparameters used across all algorithms in our framework (PPO, TRPO, A2C), including network architecture specifications and algorithm-specific parameters that we tuned to attain good performance and stable policy transfer.

Our training protocol encompassed several configurations to assess the performance of different algorithmic combinations:

- **Single algorithm training:** All configurations utilized GAE-$\lambda$ ($\lambda = 0.95$) for advantage estimation:

  – PPO: Trained for 16,000 iterations with a learning rate of $3 \times 10^{-4}$.
  – A2C: Trained for 16,000 iterations using a learning rate of $7 \times 10^{-4}$.
  – TRPO: Trained for 16,000 iterations using a learning rate of $7 \times 10^{-4}$.

- **Single pre-trained baselines:**

  – **Baseline 1 (A2C $\rightarrow$ PPO):** A2C was initially trained for 9,600 iterations, followed by PPO training for 9,600, transferring the learned policy from A2C to PPO.

**Table 5 Hyperparameters for cascading policy learning framework.**

| Parameter | PPO | TRPO | A2C |
|---|---|---|---|
| Number of iterations | 11,200 | 1,600 | 3,200 |
| Learning rate | 3e−4 | 1e−3 | 7e−4 |
| Batch size | 256 | 512 | 128 |
| **Policy network architecture** | | | |
| Hidden layers | 2 | 2 | 2 |
| Units per layer | 1,200, 1,000 | 1,200, 1,000 | 1,200, 1,000 |
| Activation function | LeakyReLU | LeakyReLU | LeakyReLU |
| Activation ($\alpha$) | 0.01 | 0.01 | 0.01 |
| **Algorithm-specific parameters** | | | |
| Discount factor ($\gamma$) | 0.99 | 0.99 | 0.99 |
| GAE parameter ($\lambda$) | 0.95 | 0.95 | 0.95 |
| Value function coef | 0.5 | – | 0.5 |
| Entropy coef | 0.01 | – | 0.01 |
| Clip range ($\varepsilon$) | 0.2 | – | – |
| Max. KL divergence | – | 0.01 | – |
| **Training configuration** | | | |
| Update steps | 2,048 | 2,048 | 128 |
| Minibatch size | 64 | 64 | – |
| Number of epochs | 10 | 10 | 1 |
| Value function updates | 10 | 10 | 1 |
| Gradient clip | 0.5 | 0.5 | 0.5 |

> – **Baseline 2 (PPO → A2C):** PPO trained for 9,600 iterations. Subsequently, A2C leverages this initial policy and further fine-tunes it for 6,000, culminating in a more stabilized policy.
>
> – **Baseline 3 (PPO → TRPO):** After an initial 9,600 iterations with PPO, TRPO continued training for 6,000 iterations, incorporating the preceding policy to achieve greater stability.

- **Double-stage cascading baselines:** These configurations investigate the cumulative effect of cascading policy knowledge through three distinct DRL algorithms to establish ideal transfer sequences. All baselines maintain consistent neural network architectures during knowledge transfer to sustain stable policy inheritance.

> – **Baseline 4 (TRPO → PPO → A2C):** TRPO trains for a specified number of iterations, then transfers its learned policy to PPO for further training. Finally, A2C fine-tunes the resulting policy, building upon the knowledge from both preceding stages.
>
> – **Baseline 5 (A2C → PPO → TRPO):** A2C initiates training and transfers its policy to PPO for intermediate optimization. PPO then transfers the refined policy to TRPO for final optimization.

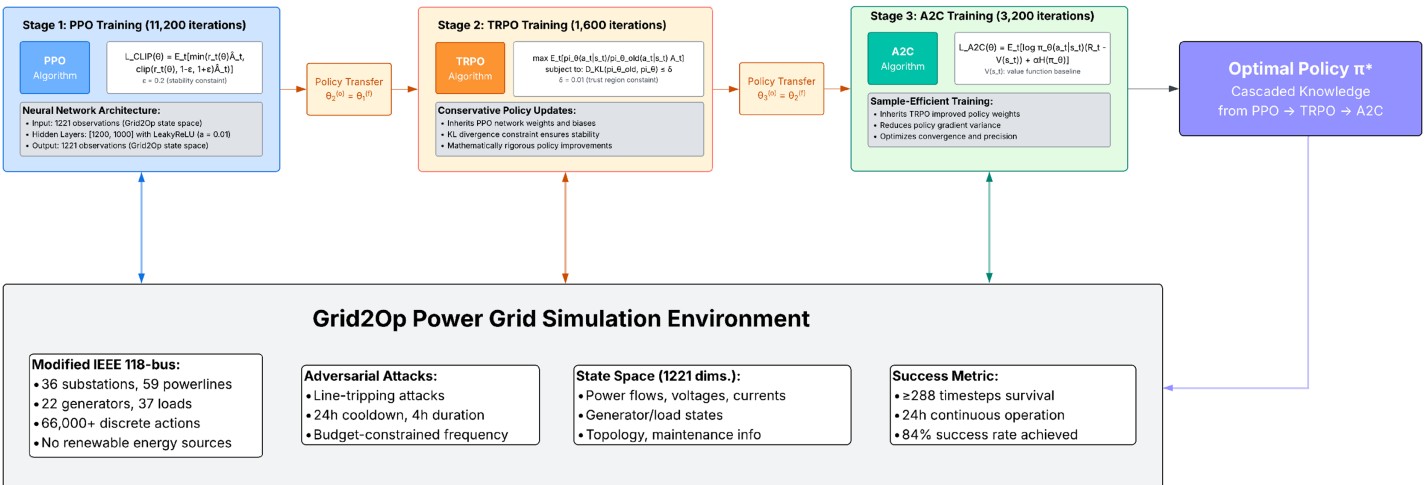

**Figure 4 Workflow of the cascading policy learning framework (CPLF) for adversarial power grid control.** The framework implements a three-stage sequential training approach: (1) proximal policy optimization (PPO) establishes stable baseline policies over 11,200 iterations through clipped surrogate objectives, (2) trust region policy optimization (TRPO) fine-tunes policies using KL-divergence constraints for mathematically rigorous updates over 1,600 iterations, and (3) advantage actor-Crtici (A2C) performs final optimization with value function baselines over 3,200 iterations. Numbered arrows represent policy-environment interactions (1, 3, 5), parameter transfer between algorithms (2, 4), and final policy deployment (6). The shared neural network architecture (two hidden layers: 1,200 and 1,000 units with LeakyReLU activation) allows for seamless knowledge transfer across all stages. The framework interfaces with the Grid2Op environment (a modified IEEE 118-bus system), featuring 36 substations, 59 transmission lines, and adversarial attack simulation.

– **Baseline 6 (PPO → A2C → TRPO):** PPO establishes an initial policy and transfers it to A2C for intermediate tuning. TRPO then performs the final optimization stage using the policy parameters from A2C.

– **Baseline 7 (A2C → TRPO → PPO):** A2C establishes an initial policy and transfers it to TRPO for intermediate refinement with monotonic improvement guarantees. PPO then performs the final optimization stage using the policy inherited from TRPO.

– **Baseline 8 (TRPO → A2C → PPO):** TRPO initiates training to ensure stable policy development, then transfers the policy to A2C for efficient intermediate training. A2C transfers the refined policy to PPO for final fine-tuning and deployment.

• **Framework training:** Our cascading framework implements a three-stage training process. Each stage inherits and builds upon the policy learned in previous stages, as in Fig. 4:

– **Stage 1:** PPO training for 9,600 iterations to establish robust baseline policies.

– **Stage 2:** TRPO optimization for 1,600 iterations, delivering conservative policy updates while maintaining performance.

– **Stage 3:** A2C fine-tuning for 3,200 iterations, focusing on sample-efficient policy improvement.

The computational requirements for CPLF implementation vary vastly across training stages. PPO training required approximately one hour on an NVIDIA Tesla P100 GPU platform for 11,200 iterations, while TRPO's conservative update mechanism completed

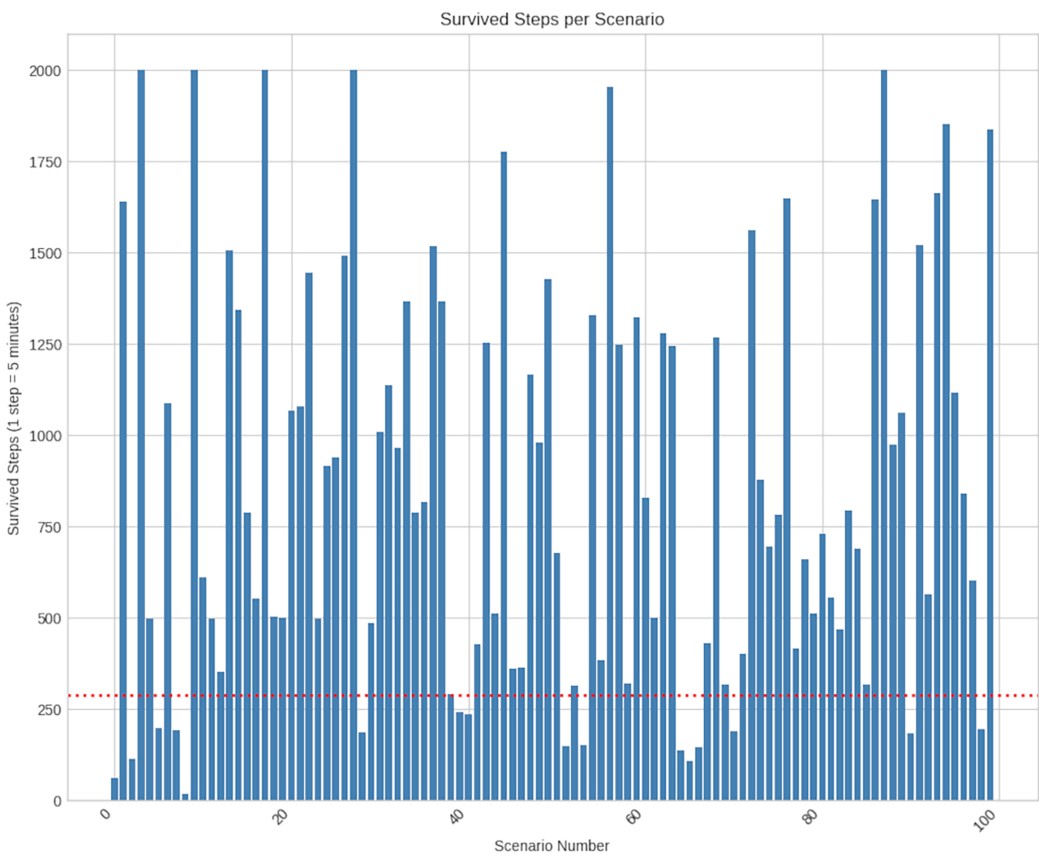

**Figure 5 Survival analysis of the cascading policy learning framework across 100 distinct adversarial test scenarios (seed 42) on the Grid2Op robustness 2020 challenge environment (modified IEEE 118-bus system).** Each point represents the total number of timesteps (0–2,000, maximum) during which the agent successfully maintained grid stability in a single scenario under bus-tripping attacks, with a 24-h cooldown and a 4-h duration. The horizontal red dashed line at 288 timesteps indicates the minimum survival threshold required for success (24 h of continuous operation at 5-min intervals). The framework achieved an 84% success rate, with 84 scenarios exceeding the threshold and many scenarios maintaining stability beyond 1,500 timesteps, demonstrating robust resilience against adversarial attacks.

1,600 iterations in 6 h. A2C fine-tuning proved the most resource-light, requiring only 1 h for 3,200 iterations due to its streamlined architecture and reduced batch size requirements.

Memory usage peaked at 16 GB during TRPO training phases, while PPO and A2C maintained more modest memory footprints of 8 and 4 GB, respectively. The training data pipeline benefits from Grid2Op's structured format, requiring minimal preprocessing overhead. State normalization occurs automatically within the environment, and the standardized observation space removes the requirement for custom feature engineering or data transformation procedures that introduce computational bottlenecks or systematic biases.

**Figure 6** **Training progression of the cascading policy learning framework showing average survival steps across the sequential three-stage training process on Grid2Op level 2 environment (seed 42, NVIDIA Tesla P100 GPU).** Stage 1 (PPO): 11,200 iterations over 1 h, achieving 107.9 mean survival steps. Stage 2 (TRPO): 1,600 iterations over 6 h, improving to 191.1 mean steps. Stage 3 (A2C): 3,200 iterations over 1 h, reaching 394.1 mean steps. The horizontal teal line at 288 timesteps represents the 24-h success threshold. Performance metrics represent rolling averages over training episodes, demonstrating a progressive knowledge accumulation through sequential algorithm deployment. Total training time: approximately 8 h.

## Performance analysis

Our empirical evaluation revealed several pronounced findings regarding the framework's performance. The survival analysis, depicted in Fig. 5, shows substantial variance across scenarios, with numerous instances exceeding 1,500 timesteps of stable operation. The visualization plots survival duration (0–2,000 timesteps) against scenario indices (0–100), with a threshold line (presented by the red line) at 288 timesteps, establishing our minimum success criterion.

To illustrate the contribution of each algorithmic stage to the CPLF's performance, we analyzed the framework's learning progression over the entire training timeline. Figure 6 illustrates this process, plotting the average number of survival steps achieved as the framework transitions through its three sequential stages.

The cascading process unfolds as follows:

1. **Stage 1: PPO Foundation (0–11,200 timesteps)** The framework begins with Proximal Policy Optimization (PPO) to establish a stable policy foundation. As shown in Fig. 6,

PPO steadily improves performance from a low baseline, creating a robust but modest initial policy that reaches an average survival of 107.9 steps. This initial phase is a prerequisite for exploring the high-dimensional action space without catastrophic policy collapses.

2. **Stage 2: TRPO (13,000–14,500 timesteps)** Upon inheriting the policy from PPO, the Trust Region Policy Optimization (TRPO) stage commences. This transition yielded an immediate and substantial performance boost. It validates our choice of TRPO, given its mathematically rigorous updates that optimize the existing policy within a constrained trust region, thereby guaranteeing monotonic improvement. This stage elevates the policy's performance to an average of 191.1 survival steps.

3. **Stage 3: A2C fine-tuning (14,500–17,500 timesteps)** In the final stage, we transfer the polished policy to the Advantage Actor-Critic agent for policy tuning. A2C's ability to use a learned value function baseline empowers more precise and sample-efficient policy improvements. The agent's performance increases in this final phase, as it decisively and consistently exceeds the 288-timestep survival target required for success, ultimately achieving an average of 394.1 survival steps.

This sequential, stage-wise visualization confirms that the CPLF's superior performance is not incidental but a direct result of its cascading design. As illustrated in the detailed workflow presented in Fig. 4, each algorithm builds upon the distilled knowledge of its predecessor, validating our hypothesis that sequential knowledge transfer across complementary algorithms yields a more robust control policy for power grid resilience. The subsequent analysis compares the final performance of this complete framework against a range of baseline configurations across 100 adversarial scenarios. Due to computational constraints ($\approx$8 h per training run on NVIDIA P100 GPUs), we trained each permutation once; however, we evaluated each resulting policy across 100 diverse random adversarial scenarios (seeds 300–399) to ensure statistical robustness of our performance comparisons.

To quantify the individual contributions of each algorithmic component, we conducted a complete ablation study examining the compelling performance hierarchy across different baselines. As depicted in Fig. 7, each data point represents the mean survival duration (0–2,000 timesteps) over 100 scenarios. Our rationale for the PPO $\rightarrow$ TRPO $\rightarrow$ A2C sequence is grounded in algorithmic complementarity: PPO's clipped surrogate objective fosters stable initial policy development, which is a prerequisite for success in high-dimensional power grid environments, TRPO's KL divergence constraint guarantees mathematically rigorous updates by preserving well-established policy patterns, and A2C's value function baseline achieves precise policy improvements through reduced gradient variance. The results reveal clear evidence for the necessity of each element in achieving optimal performance.

Single-algorithm baselines revealed severe limitations in managing adversarial power grid scenarios. The Random Agent baseline and standalone PPO both achieved 0% success rates, failing to maintain grid stability beyond the minimum threshold in any test scenario. PPO's failure stems from its exploration strategy, which proves inadequate for the highly

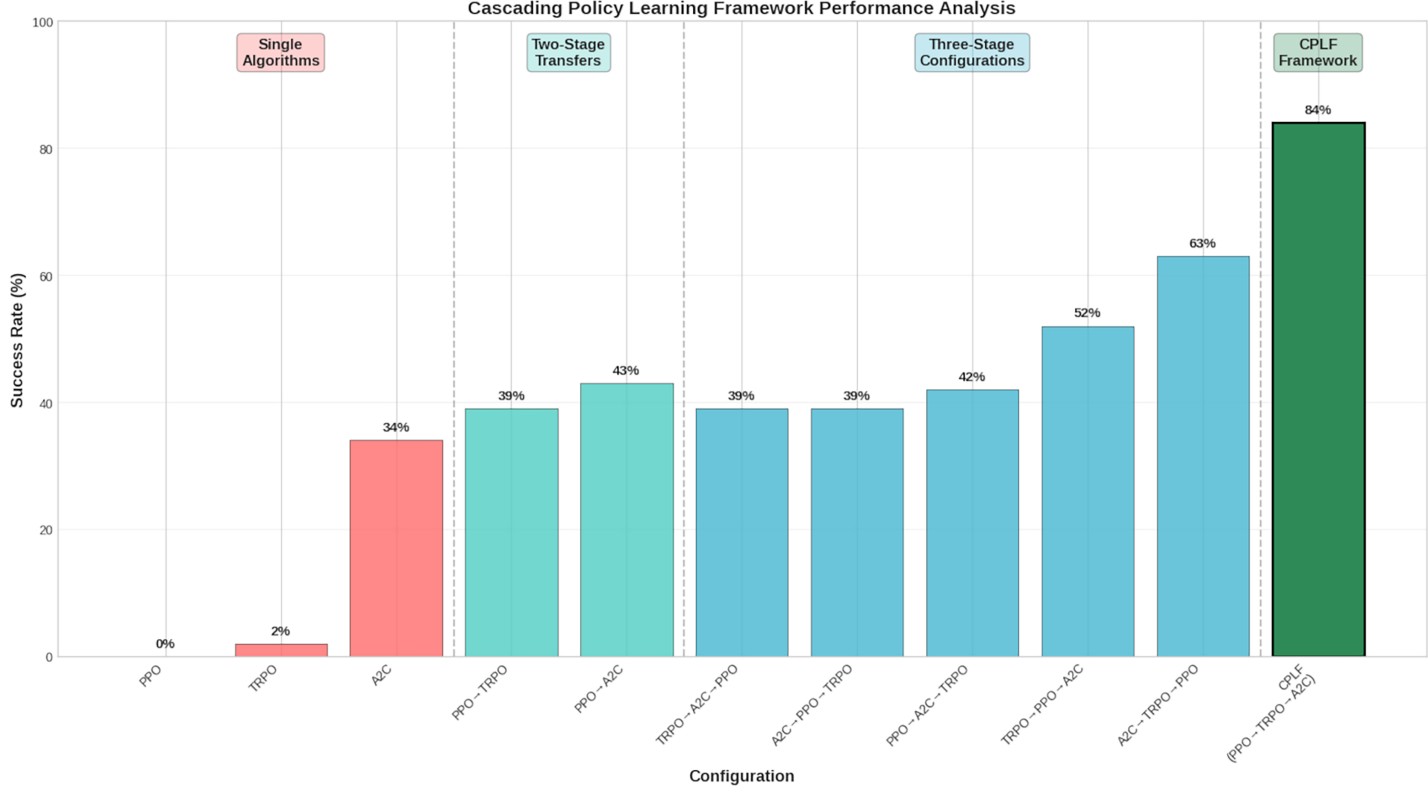

**Figure 7 Comparative performance analysis of the proposed cascading policy learning framework (CPLF) against baseline reinforcement learning agents across different architectural configurations, where each configuration was evaluated over 100 episodes (seed 42) in a Grid2Op power grid management environment under adversarial conditions with episodes lasting up to 2,000 timesteps.** The success rate is defined as the percentage of episodes in which an agent maintained grid stability for at least 288 timesteps (24 h) without experiencing a blackout. The CPLF achieves an 84% success rate, significantly outperforming single-algorithm agents including proximal policy optimization (PPO: 0%), trust region policy optimization (TRPO: 2%), and advantage actor-critic (A2C: 34%), two-stage transfers (PPO → TRPO: 39%, PPO → A2C: 43%, A2C → PPO: 43%), and three-stage configurations (TRPO → A2C → PPO: 39%, TRPO → PPO → A2C: 40%, A2C → PPO → TRPO: 39%, PPO → A2C → TRPO: 62%, A2C → TRPO → PPO: 63%). The progressive improvement from single-algorithm baselines (max: 34%) to two-stage transfers (max: 43%) to three-stage configurations (max: 63%) and finally to the optimized CPLF (84%) demonstrates that sequential knowledge transfer across complementary algorithms with optimized ordering yields superior control policies for power grid resilience.

constrained and safety-critical power grid environment. A standalone TRPO achieved only a 2% success rate (average survival: 34.6 steps, median: 27.0 steps), demonstrating that conservative policy updates alone are insufficient without a stable initial foundation. Standalone A2C performed moderately better with a 34% success rate, benefiting from its value function baseline but still lacking the robustness required for consistent performance.

The introduction of policy transfer between algorithms resulted in meaningful improvements, validating the utility of knowledge transfer between complementary algorithms. The PPO → TRPO configuration achieved a 39% success rate, with an average survival of 462.7 steps and a median of 193.0 steps, suggesting that TRPO's conservative optimization can build upon PPO's foundation but remains limited without subsequent fine-tuning. The PPO → A2C configuration reached a 43% success rate (average survival:

469.2 steps, median: 205.5 steps), confirming the benefit of PPO's initial policy establishment, followed by A2C's optimized fine-tuning capabilities.

Alternative three-stage configurations further illustrated the imperative of proper algorithmic sequencing. The TRPO → PPO → A2C sequence achieved a 52% success rate, with an average survival of 418.5 steps and a median of 319.0 steps. This reduction stemmed from the TRPO's conservative nature and proved inadequate for initial policy exploration in the high-dimensional Grid2Op environment. A2C → PPO → TRPO reached a 33% success rate (average survival: 350.8 steps, median: 98.5 steps), because the unstable foundations created by A2C's high-variance initial learning hindered subsequent algorithms from achieving stability. PPO → A2C → TRPO achieved a 42% success rate (average survival: 404.3 steps, median: 111.5 steps), with TRPO's conservative adjustments in the final position unable to fully capitalize on the A2C intermediate stage. Table 6 provides a detailed statistical analysis, including the mean, standard deviation, median, 95th percentile, coefficient of variation, and performance stability metrics for all multi-stage configurations (statistics achieved by our solution, CPLF, are shown in bold). These results confirm that initiating the cascade with PPO's stable foundation is a prerequisite and that TRPO's conservative optimization works best as an intermediate stage rather than as a final component.

The complete CPLF framework, which extends the knowledge transfer concept by incorporating TRPO as an intermediate optimization stage, achieved an exceptional 84% success rate with an average survival of 861.0 steps and a median of 786.0 steps. As shown in Fig. 8, the framework's robust performance is further evidenced by its superior survival duration distribution, with the highest median survival and most consistent performance across all scenarios. The framework's 95th percentile survival duration of 1,957.2 steps reflects consistent high performance across challenging scenarios, distinguishing it from the more variable performance patterns exhibited by other configurations. This framework leverages TRPO's ability to deliver mathematically rigorous updates, building upon the initial policy established by PPO and establishing a strong foundation for final tuning by A2C. The 41-percentage-point improvement over the best two-stage configuration (PPO → A2C: 43%) reveals that the additional architectural complexity yields meaningful advantages in grid stability management. The progressive performance improvements from single-algorithm (0–34%) to two-stage transfers (39–43%) and finally to the complete CPLF framework (84%) substantiate the core premise of our cascading approach: that sequential knowledge transfer across multiple algorithmic stages delivers markedly superior policy development compared to simpler architectural configurations.

Figure 9 illustrates the relationship between success rate and survival duration across all configurations, with the complete quantitative results detailed in Table 7. Our CPLF framework uniquely occupies the optimal performance quadrant, achieving both a high 84% success rate and an extended 861-timestep survival duration, thereby establishing a new benchmark for power grid resilience under adversarial conditions. While most other configurations exhibit clear trade-offs between these metrics, the progressive performance improvements from single-algorithm (0–34%) to two-stage transfers (39–43%) and finally to the complete CPLF framework (84%) substantiate the pivotal premise of our cascading

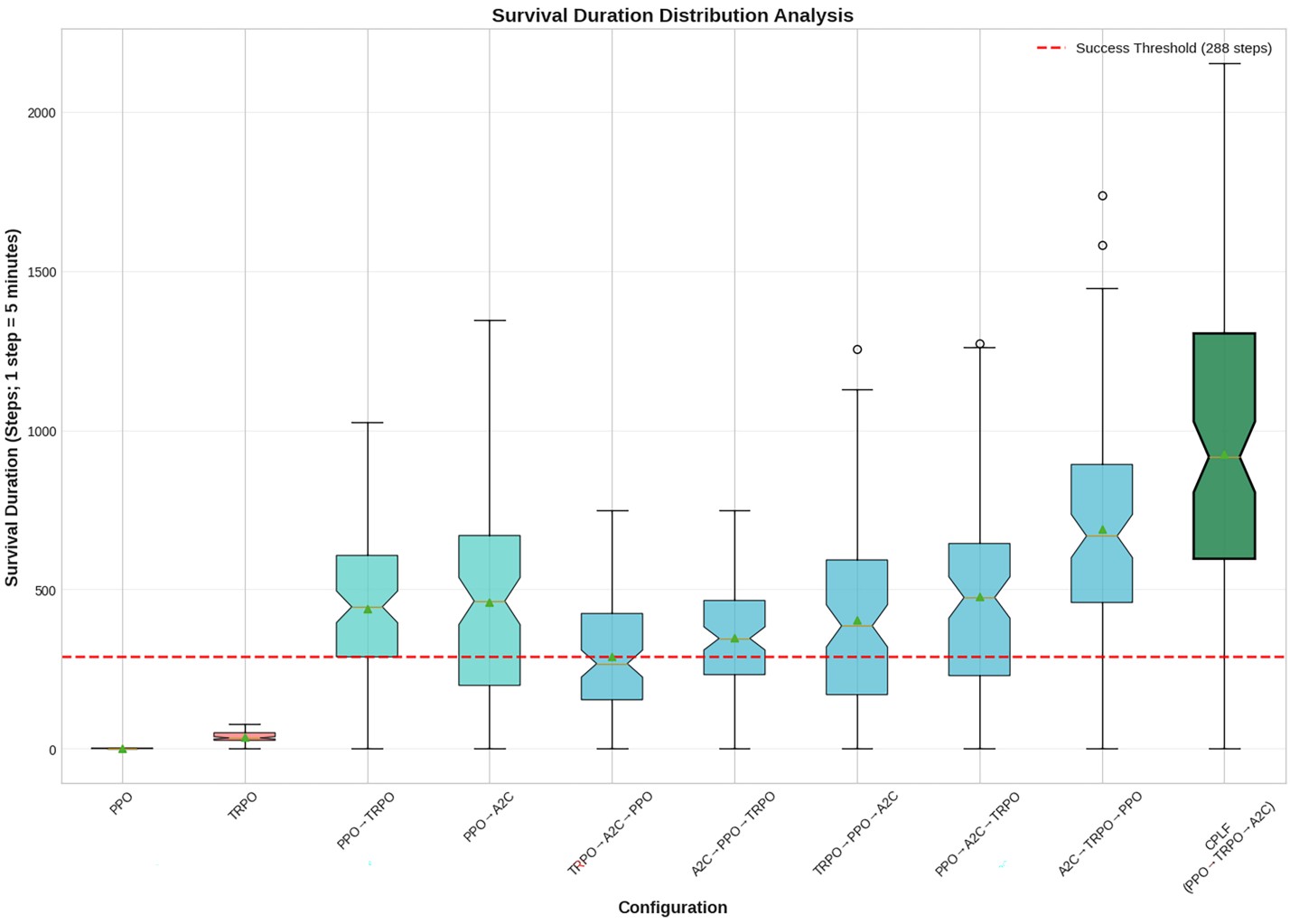

**Figure 8 Survival duration distribution analysis across different reinforcement learning configurations evaluated on 100 adversarial test scenarios (seed 42) in the Grid2Op Robustness 2020 challenge environment.** Box plots display survival time distributions in timesteps (y-axis, 0–2,000 range), where the central line represents median survival, box boundaries indicate $25^{th}$ and $75^{th}$ percentiles (IQR), and whiskers extend to $1.5 \times$ IQR. The horizontal red dashed line marks the 288-timestep success threshold (24 h). The cascading policy learning framework (CPLF) achieves superior performance with median = 786 steps, mean = 861 steps, and $95^{th}$ percentile = 1,957 steps. Configurations tested include: single algorithms (TRPO median = 27), two-stage transfers (PPO → TRPO median = 193, PPO → A2C median = 205), and three-stage configurations (TRPO → PPO → A2C median = 319, A2C → PPO → TRPO median = 98, PPO → A2C → TRPO median = 111). Here, PPO stands for proximal policy optimization, TRPO for trust region policy optimization, and A2C for advantage actor-critic.

approach: sequential knowledge transfer across multiple algorithmic stages delivers markedly superior policy development compared to simpler architectural configurations.

Our experimental design mitigates the risk of overfitting by training on "level 2" scenarios and evaluating 100 unseen, more challenging "competition level" scenarios. The framework's 84% success rate on this holdout set affirms strong generalization. Training stability stemmed from the inherent properties of the cascaded algorithms (PPO's clipping, TRPO's KL-divergence constraint, and A2C's variance reduction). The monotonic performance improvement and smooth convergence shown in Fig. 6 corroborate the successful implementation of this approach.

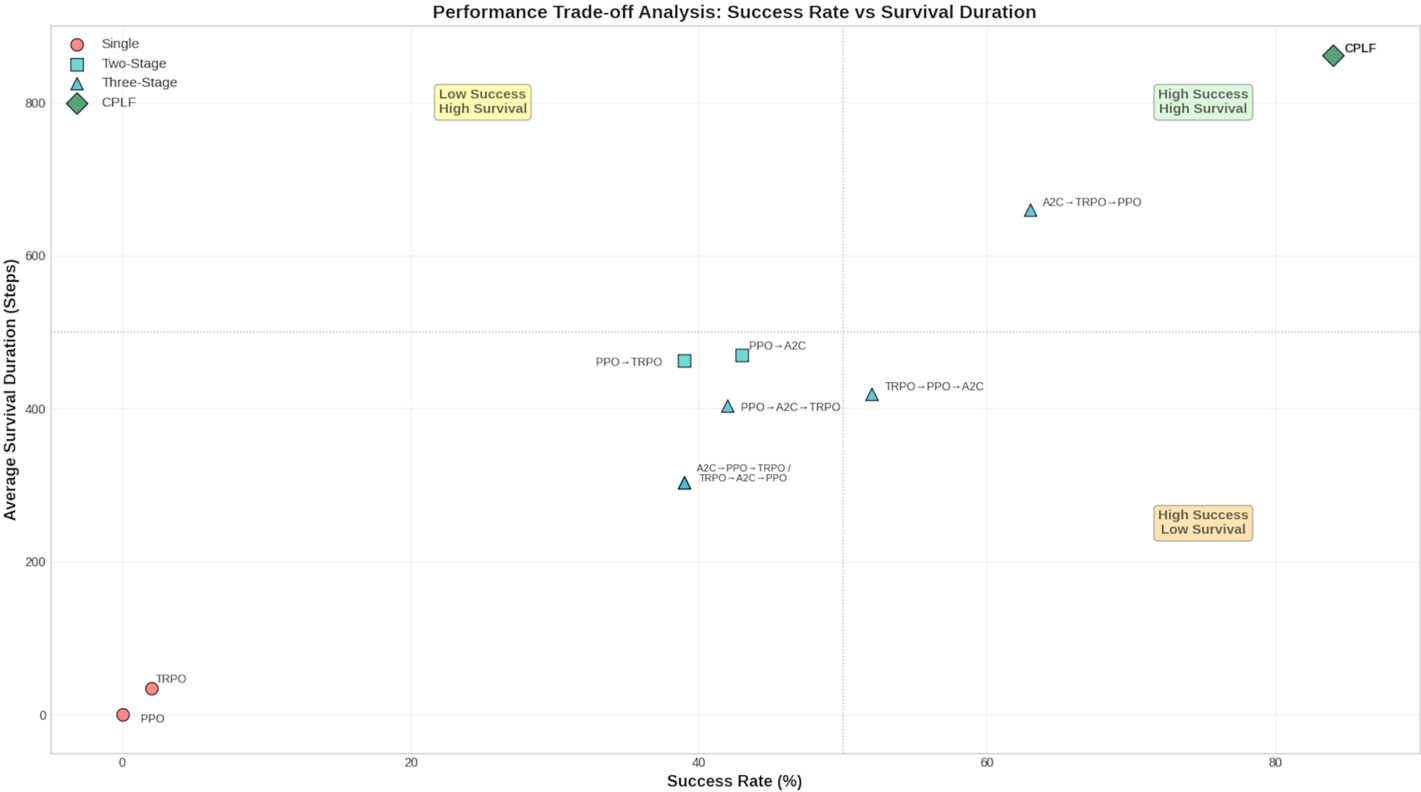

**Figure 9 Performance trade-off analysis illustrating the relationship between success rate (%, x-axis) and average survival duration (timesteps, y-axis) across all tested configurations evaluated on 100 adversarial scenarios (seed 42, Grid2Op Robustness 2020 challenge).** Each marker represents a distinct configuration: single algorithms (red circles), two-stage transfers (teal squares), three-stage configurations (blue triangles), and the CPLF framework (green diamond). The success rate is defined as the percentage of episodes surviving $\geq 288$ timesteps (24 h). The CPLF uniquely occupies the optimal performance quadrant at (84%, 861 timesteps), outperforming all baselines. Quadrant divisions highlight performance regions where most configurations exhibit trade-offs between metrics. Abbreviations: CPLF, cascading policy learning framework; PPO, proximal policy optimization; TRPO, trust region policy optimization; A2C, advantage actor-critic. The box plot for CPLF (far right) clearly shows a superior distribution, achieving a median survival of 786.0 steps and a 75th percentile of over 1,200 steps, demonstrating its consistent high performance compared to all other configurations, whose medians remained below 500 steps.

While the 84% success rate demonstrates strong overall performance, analyzing the 16 failures provides valuable insights into the framework's operational boundaries and remaining vulnerabilities. To complement the success rate metric and provide deeper insights into resilience characteristics, we conducted a failure analysis examining constraint violations across all unsuccessful episodes. This analysis reveals two primary failure patterns that account for all 16 constraint violations (16 failed scenarios out of 100 total). The dominant failure mode, occurring in 10 of these cases, involved concurrent high-impact contingencies where the adversary strategically initiated an attack on a critical transmission line immediately followed by a scheduled maintenance outage on a nearby parallel line. This $N-2$ situation created a severe topological bottleneck that the agent could not resolve sufficiently quickly, ultimately leading to cascading overloads. The remaining six failures exhibited a distinct pattern characterized by substation isolation, where coordinated attacks on multiple lines connected to a single substation effectively severed it from the central grid, resulting in immediate blackouts for the loads it served.

**Table 6 Quantitative performance summary of all evaluated configurations.** Values in bold represent the statistics achieved by our solution.

| Configuration | Success rate (%) | Avg. survival (steps) | Median survival (steps) | 95th percentile (steps) |
|---|---|---|---|---|
| PPO | 0.0 | 0.0 | 0.0 | 0.0 |
| TRPO | 2.0 | 34.6 | 27.0 | 51.1 |
| PPO → TRPO | 39.0 | 462.7 | 193.0 | 1,541.8 |
| PPO → A2C | 43.0 | 469.2 | 205.5 | 1,628.8 |
| A2C → PPO → TRPO | 39.0 | 303.9 | 236.0 | 681.4 |
| TRPO → A2C → PPO | 39.0 | 303.9 | 236.0 | 681.4 |
| PPO → A2C → TRPO | 42.0 | 404.3 | 111.5 | 1,349.3 |
| TRPO → PPO → A2C | 52.0 | 418.5 | 319.0 | 1,249.8 |
| A2C → TRPO → PPO | 63.0 | 659.4 | 494.0 | 1,579.0 |
| **CPLF** | **84.0** | **861.0** | **786.0** | **1,957.2** |

**Table 7 Performance statistics for multi-stage configurations.** Values in bold represent the statistics achieved by our solution.

| Configuration | Success rate | Mean ± Std | Median | 95th %ile | CV | Perf. stability |
|---|---|---|---|---|---|---|
| **CPLF** | **84.0%** | **861.0 ± 597.1** | **786.0** | **1,957.2** | **0.72** | **0.87** |
| A2C → TRPO → PPO | 63.0% | 659.4 ± 560.1 | 494.0 | 1,579.0 | 0.85 | 0.75 |
| TRPO → PPO → A2C | 52.0% | 418.5 ± 328.7 | 319.0 | 1,249.8 | 0.79 | – |
| PPO → A2C → TRPO | 42.0% | 404.3 ± 344.7 | 111.5 | 1,349.3 | 0.85 | – |
| A2C → PPO → TRPO | 39.0% | 303.9 ± 163.0 | 236.0 | 681.4 | 0.54 | 0.78 |
| TRPO → A2C → PPO | 39.0% | 303.9 ± 163.0 | 236.0 | 681.4 | 0.54 | 0.78 |

## DISCUSSION

By merging PPO, A2C, and TRPO into a unified learning system, the CPLF framework introduces a novel approach to power grid control. This methodology promotes robust policy learning and the seamless sharing of strategies, ultimately delivering a more resilient grid. Experimental results confirm that this multi-stage strategy conspicuously outperforms DRL baselines in fortifying the grid against malicious attacks.

While our approach exemplifies promising results, several limitations warrant discussion. The framework operates in a high-dimensional action space, comprising more than 66,000 possible actions, which is computationally intensive and may limit real-time applicability in resource-constrained environments. Additionally, the framework's performance exhibits sensitivity to hyperparameter selection for the three agents, including learning rate, batch size, and policy transfer timing, necessitating rigorous fine-tuning that can be time-consuming and demand domain expertise. Furthermore, our evaluation is currently limited to the modified IEEE-118 bus system, and scalability to large-scale power grids remains to be validated. While our evaluation procedure across 100 scenarios with varied seeds provides evidence of generalization within the tested distribution, systematic testing across controlled distribution shifts (*e.g.*, extreme seasonal profiles, multi-line outages, specific unseen contingencies) would strengthen external validity claims and provide deeper insights into the boundaries of policy robustness.

Specifically, while our framework generalized across randomized attack scenarios and unseen competition-level environments, it has not yet been systematically tested on multi-line outages, seasonal load variations, or domain randomization setups. We acknowledge this as a limitation and identify it as a priority for future work.

A notable limitation of our evaluation is the absence of direct comparison with non-RL baselines such as OPF/SCOPF-based redispatch, rule-based priority shedding, or MPC approaches. While the Grid2Op platform provides baseline agents (RandomAgent, ExpertAgent), implementing fair comparisons with traditional power systems control methods requires extensive engineering effort due to different problem formulations and the platform's primary design for DRL evaluation. It hinders a full contextualization of the performance gains relative to classical control paradigms. We recognize that comparisons with OPF/SCOPF and rule-based heuristics are critical for bridging the DRL–power systems gap. While implementing these within Grid2Op was beyond our current revision scope, we explicitly identify this as essential future work. The framework's robustness against more evolved attack models beyond the scenarios tested also mandates further investigation.

Nevertheless, several avenues for future research present promising opportunities to bolster the practical application and applicability of CPLF. Future work should incorporate domain randomization during training and structured out-of-distribution testing protocols to comprehensively assess policy robustness across diverse operational regimes. Establishing rigorous benchmarks against classical power systems control methods (OPF/ SCOPF, MPC, rule-based strategies) represents a critical priority to fully quantify the advantages of cascading DRL methods over traditional control paradigms in adversarial scenarios. Such comparisons would help validate whether the additional complexity of multi-stage DRL training yields proportional benefits compared to well-tuned classical approaches.

Reducing the action space to include only the most meaningful and critical actions, as suggested by Binbinchen's work (*Marot et al., 2021*), could streamline the learning process and accelerate policy convergence. This approach is similar to focusing on electrical isolation between object groups in substation topologies rather than specific busbar assignments.

Additionally, incorporating knowledge distillation techniques during the transfer of policies between agents could further boost performance. Knowledge distillation, a paradigm first proposed in *Hinton (2015)*, trains a smaller, more lightweight student model to replicate the performance of a larger, higher-capacity teacher model. This process reduces computational cost while preserving high accuracy. We can achieve smoother and more superior policy transfers by distilling knowledge from pre-trained models.

Furthermore, to achieve even better performance, we will increase the number of learning iterations and conduct more extensive hyperparameter tuning. Precisely, we will investigate automated hyperparameter optimization using Bayesian frameworks such as Optuna. This approach promises to accelerate convergence and reduce computational overhead. It has shown promising results over traditional grid and random search methods in reinforcement learning applications (*Akiba et al., 2019*). Integrating wide-ranging

AutoML techniques, including automated neural architecture search, could further streamline the framework's deployment across a diverse set of power grid topologies by identifying optimal network topologies and eliminating the current manual optimization overhead for each new scenario.

Addressing safety constraints represents another critical direction for future work. While our current framework demonstrates robust performance through its cascaded architecture, incorporating formal safe RL techniques would provide explicit safety guarantees during both training and deployment. Methods such as constrained policy optimization or safety layers could ensure that the agent maintains critical operational constraints even during exploration, which is particularly important for critical infrastructure applications where constraint violations can have severe consequences.

Moreover, we will focus on more powerful opponent attacks by increasing both the attack complexity (*e.g.*, tripping multiple lines per timestep) and the opponent's total attack budget. Reducing the time between two successive attacks and increasing the downtime period for each attack can considerably amplify the challenge the opponent encounters. It will guarantee that our control policies are resilient against diverse malevolent scenarios.

Scaling up to large-scale power grids is imperative to validate the framework's scalability. Developing solutions for modern power grids that integrate renewable energy sources will also be critical for overcoming real-world challenges in sustainable energy management.

Lastly, we will investigate the integration of other frameworks that complement our cascading approach. Federated RL could support distributed training across multiple grid regions while preserving the sequential knowledge transfer within each federated node. Hierarchical RL (HRL) could decompose our framework into high-level policy selection (choosing between PPO/TRPO/A2C strategies) and low-level action execution (specific grid operations) (*Manczak, Viebahn & van Hoof, 2023*; *Jendoubi & Bouffard, 2023*; *Hutsebaut-Buysse, Mets & Latré, 2022*; *Narvekar et al., 2020*; *Matavalam et al., 2022*). Curriculum RL (CRL) systematically orders the cascading stages by progressively increasing scenario complexity, starting with simple grid contingencies for PPO, advancing to multi-fault scenarios for TRPO, and culminating in adversarial attacks for A2C. These structured learning paradigms could optimize the reliability and adaptability of our sequential knowledge transfer approach.

## CONCLUSION

The Cascading Policy Learning Framework uses a novel sequential training approach to fortify power grid resilience and optimize control policy expediency. Our framework, which operates within a modified IEEE-118 power network, leverages the complementary strengths of multiple DRL algorithms, including Proximal Policy Optimization, Advantage Actor-Critic, and Trust Region Policy Optimization. This integration combines stability with PPO's constrained updates, resilience with TRPO's KL-divergence constraint, and precision with A2C's value-function baseline. These features allow the policy to adapt to

unstable grid scenarios, including rapid load changes and targeted attacks. Our solution achieved an 84% success rate in test scenarios, maintaining grid stability for at least 24 h and substantially outperforming a plethora of baseline agents. It manifests the capability of our multi-stage learning strategy in strengthening power grid resilience against malicious attacks. By integrating these algorithms, CPLF overcomes limitations in existing approaches, particularly in adapting to hostile conditions and maintaining operational continuity. This work establishes a blueprint for applying sequential learning and multi-stage RL strategies to large-scale control problems in domains such as transportation networks, water distribution systems, and telecommunications infrastructure, where resilience is paramount. Moreover, using other frameworks, such as curriculum reinforcement learning and hierarchical RL, could unlock promising perspectives for improving the current results.

## CODE AVAILABILITY

The complete source code for the Cascading Policy Learning Framework, including implementation details, hyperparameter configurations, and evaluation scripts, has been made publicly available through a permanent DOI-backed archive (DOI: 10.17605/OSF. IO/GCW7X) (*Bensalah et al., 2025*). The repository includes detailed documentation, installation instructions, and complete reproducibility guidelines to support replication and extension of this work. We provide all experimental configurations to ensure full reproducibility of reported results, including an interactive Jupyter notebook with the environment configuration. The requirements.txt file documents the exact version dependencies of the used packages, such as Python 3.11.13, Grid2Op 1.12.1, lightsim2grid 0.10.3, and PyTorch 2.6.0+cu124. The interactive Jupyter notebook provides the cascading policy learning framework along with the environment configuration.

### Funding

This work was supported by the Directorate General for Scientific Research and Technological Development (DGRSDT), Ministry of Higher Education and Scientific Research, Algeria, through its support to the research projects of the Research Laboratory on Computer Science's Complex Systems (ReLa(CS)$^2$). It has also been supported by the French Government under the "France 2030" program, as part of the IRT SystemX Programme CYBELIA, and the European Union's Horizon Europe research and innovation programme DYNAMO, under grant agreement no. 101069601. The funders had no role in study design, data collection and analysis, decision to publish, or preparation of the manuscript.

### Grant Disclosures

The following grant information was disclosed by the authors:
Directorate General for Scientific Research and Technological Development (DGRSDT).
Ministry of Higher Education and Scientific Research, Algeria.

Research Laboratory on Computer Science's Complex Systems (ReLa(CS)[2]).

French Government under the "France 2030" program, as part of the IRT SystemX Programme CYBELIA.

European Union's Horizon Europe research and innovation programme DYNAMO: 101069601.

## Competing Interests

Reda YAICH is employed by IRT SystemX, a non-profit private research institute based in Palaiseau, France. The authors declare that there are no other competing interests.

## Author Contributions

- Assem Sohaib Bensalah conceived and designed the experiments, performed the experiments, analyzed the data, performed the computation work, prepared figures and/or tables, authored or reviewed drafts of the article, and approved the final draft.
- Toufik Marir conceived and designed the experiments, performed the experiments, analyzed the data, prepared figures and/or tables, authored or reviewed drafts of the article, and approved the final draft.
- Reda Yaich conceived and designed the experiments, performed the experiments, analyzed the data, authored or reviewed drafts of the article, and approved the final draft.
- Mohamed Sedik Chebout performed the experiments, analyzed the data, authored or reviewed drafts of the article, and approved the final draft.

## Data Availability

The code is available in the Supplemental Files.

The code is available at OSF: BENSALAH, A. S., Marir, T., Yaich, R., & CHEBOUT, M. S. (2025, October 3). Cascading Policy Learning Framework (CPLF). https://doi.org/10.17605/OSF.IO/GCW7X.

## Supplemental Information

Supplemental information for this article can be found online at http://dx.doi.org/10.7717/peerj-cs.3358#supplemental-information.

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
