# Peer review of "A cascading policy learning framework for enhancing power grid resilience"

_PeerJ Computer Science, doi:10.7717/peerj-cs.3358_

## Round 0.1 · original submission · Major Revisions

· Academic Editor

Major Revisions

Reviewer 1 ·

Basic reporting

While the use of three sequential DRL agents is novel, the paper does not sufficiently compare or justify why the chosen sequence (PPO → TRPO → A2C) is optimal. Would TRPO → PPO → A2C or other sequences perform worse or better?

The theoretical underpinning of the cascading knowledge transfer is not deeply analyzed. Could a transfer learning or meta-learning explanation strengthen the scientific novelty?

The manuscript lacks details about how policy transfer is conducted between stages. Is it weight initialization only? Are there any freezing or regularization mechanisms?

The scenario generation methodology (e.g., type and pattern of adversarial attacks) needs more description. Are these randomized, rule-based, or scenario-driven?

It is unclear whether performance gains are statistically significant. Confidence intervals or standard deviations are not reported.

Experimental design

none

Validity of the findings

Include additional performance metrics to enrich the analysis.

Provide better captions and labels for figures.

Discuss potential overfitting or stability concerns in the training process.

Additional comments

Revise grammatical issues for smoother readability.

Minor Comments
Line 266: Clarify how 288 timesteps is a valid success criterion—cite prior work or explain threshold rationale.

Table 3: Use consistent notation for parameters (e.g., γ for discount factor, λ for GAE).

Line 303: Use “teacher-student” terminology more formally, referencing standard knowledge distillation frameworks.

Line 324: State whether the simulation environment includes renewable sources or only traditional generation.

Line 337: Consider discussing how parameter sensitivity could be mitigated using AutoML or Bayesian optimization.

Cite this review as

Reviewer 2 ·

Basic reporting

The article is generally well written, and the language used is clear and professional. The introduction presents the motivation adequately, situating the study within the context of cyber-physical systems and the challenges of maintaining power grid resilience under adversarial threats. The background references are relevant and sufficient to contextualize the proposed approach, though the literature review could be enhanced with a deeper comparison to other multi-agent or hierarchical DRL approaches.

The structure of the article conforms to PeerJ norms. However, there are some minor grammatical repetitions and slightly awkward phrasing in the abstract and main body (e.g., repetitive use of “refines it with…”). These do not impede comprehension but should be revised for clarity.

There is no formal theorem or proof-based component, which is acceptable for this type of applied AI paper. Nonetheless, the paper would benefit from clearly defining key terms such as "adversarial attacks" in the context of CPS and explicitly describing the baseline models for comparison.

Suggested improvements:

Revise redundant phrasing in the abstract and clarify the cascading structure.

Consider adding a figure illustrating the workflow of the proposed framework.

Experimental design

The study aligns well with the journal’s scope. The investigation is scientifically valid and falls within current trends in the use of DRL for infrastructure resilience. The use of PPO, TRPO, and A2C in a sequential manner is innovative and technically sound.

However, the experimental design lacks sufficient detail for replication:

The simulation environment is not adequately described. Details such as grid topology, simulation parameters, load variation, and attack patterns are necessary for reproducibility.

Evaluation methods and assessment metrics (e.g., how the "84% success rate" is calculated) are not clearly explained.

There is no discussion of data preprocessing or system constraints (e.g., latency, training time, hardware resources).

Suggested improvements:

Provide more comprehensive details of the simulation environment and attack settings.

Include a description of the DRL agent architectures and hyperparameters.

Mention the availability of source code or consider releasing it for transparency and reproducibility.

Validity of the findings

The paper presents promising empirical results, demonstrating that the cascading framework achieves improved performance over baseline agents in resisting bus-tripping attacks. The conclusions are consistent with the reported results, and the argument aligns with the goals stated in the introduction.

However, the findings would be more convincing if supported by:

An ablation study that quantifies the contribution of each component (PPO, TRPO, A2C).

A discussion of limitations, such as computational complexity, scalability to larger grids, or sensitivity to different attack models.

Insights into why the cascading structure yields superior performance, possibly with learning curves or gradient behavior comparisons.

Suggested improvements:

Add a brief ablation analysis (or discussion) of individual DRL components.

Acknowledge and discuss limitations and future work directions more explicitly.

Additional comments

The paper is well-motivated and presents a novel and effective approach to a relevant real-world problem. The cascading integration of DRL algorithms is a valuable contribution, particularly for domains where stability and robustness are crucial.

However, the paper would benefit from improved clarity in the experimental section, more technical details, and a visual aid to illustrate the overall workflow.

Cite this review as

---

## Round 0.2 · Minor Revisions

· Academic Editor

Minor Revisions

Dear Authors, thank you for the revised submission.

Some concerns remain, please take into account the reviewer's suggestion and clarify their doubts. I am confident that your proposal can be greatly improved.

Reviewer 1 ·

Basic reporting

-

Experimental design

-

Validity of the findings

-

Cite this review as

Reviewer 3 ·

Basic reporting

Overall clarity & organization.
The manuscript is generally clear and well structured, with a logical flow from motivation to results. The prose is understandable, but several passages would benefit from tightening and from earlier definitions of key concepts.

Actionable suggestions

Define terms up front. Provide a concise operational definition of resilience vs robustness early in the Introduction and use the terms consistently throughout.
Make the contribution statement explicit. End the Introduction with a bulleted list of contributions (method, evaluation, open-source release) tied to specific sections.

- Related work focus. The survey is broad; emphasize comparability. For each family (RL, optimal power flow/MPC, game-theoretic defense), state what gap remains and how CPLF addresses it.
- Figures & captions. Ensure captions are self-contained (experiment conditions, metrics, units, #seeds, #episodes). Add axis units and abbreviations expansions. Consider adding learning-curve plots and boxplots with variance.
- Tables to add.
(i) A complete hyperparameter table for PPO/TRPO/A2C;
(ii) An environment/configuration table (Grid2Op version, backend, IEEE-118 modifications, obs/action spaces, reward weights);
(iii) A KPI table that defines all metrics (survival time, load shed, limit violations, interventions).
- Style & language. Minor issues (capitalization of headings, occasional extra spaces/commas). A careful language pass will improve polish.
- Data/code availability. Since reproducibility is central to PeerJ, please provide a permanent link (e.g., DOI-backed archive) and cite it in-text; pin all versions (Python, Grid2Op/lightsim2grid, DL framework, CUDA).

Experimental design

Suitability & originality.

The cascading sequence PPO → TRPO → A2C is an interesting idea that plausibly leverages complementary strengths. The research question is clear and suitable for PeerJ’s scope.

Actionable suggestions:

- Justify the cascade order. Add an ablation over all 6 permutations of the three algorithms (≥5 random seeds) and discuss why the proposed order is preferable.
- Stage-transition criteria. Specify stopping/transition rules (fixed steps, performance plateau, KL threshold). Clarify which parameters are transferred across stages (actor, critic, both).
- Baselines beyond RL. Include non-RL comparators commonly used in power-systems studies: OPF/SCOPF-based redispatch heuristics, rule-based agents (priority shedding/line switching), and a simple MPC/greedy look-ahead.
- Safety & constraints. Detail how thermal/voltage/N-1 constraints are treated in training and evaluation (hard action masking, penalties, safety layer). Provide the explicit reward function with weights and rationale.
- Scenario design. Describe attack/disturbance generation: types (line/generator outages, sensor spoofing), frequency, duration, and whether selection is random or adversarial.
- Generalization tests. Evaluate under distribution shift: unseen contingencies, different load/generation regimes, seasonal profiles, and multi-line outages; consider domain randomization.
- Compute budget & sample efficiency. Report wall-clock training time, environment steps, steps/sec, GPU/CPU specs per stage; show learning curves and early-stopping behavior.
- Hyperparameter search. Briefly document the tuning procedure (grid/random/Optuna), ranges, and the number of trials to avoid cherry-picking.

Validity of the findings

Current evidence.
Results indicate improved performance (~84% success in maintaining service continuity) and appear consistent across tested scenarios. However, the strength of the claims would improve with broader baselines, multiple KPIs, and statistical analysis.

Actionable suggestions
- Statistical reporting. For each metric, report mean ± std (or 95% CI) across seeds and scenarios; include significance tests (e.g., paired Wilcoxon) against each baseline.
- Multiple KPIs. Complement “success rate” with load shed, number/duration of constraint violations, number of interventions, and recovery time to better capture resilience.
- Fair comparisons. Ensure all baselines receive comparable training budgets and tuning effort; state any deviations.
- Failure analysis. Characterize the ≈16% failures (typical cascade patterns, bottleneck components) and provide qualitative examples/plots.
- External validity. Discuss how results might transfer to larger topologies (e.g., IEEE-300) or to mixed cyber-physical threats; clearly acknowledge simulation-only limitations.

Additional comments

The manuscript addresses an important problem with a creative DRL pipeline and is written with commendable clarity. To reach publishable quality, I recommend focusing the revision on: (i) a principled justification via permutation ablations, (ii) inclusion of non-RL baselines to contextualize gains, (iii) explicit safety/reward design and constraint handling, (iv) full compute/sample-efficiency reporting, and (v) comprehensive reproducibility materials (pinned versions, seeds, HP table, and an archival code release). With these enhancements, the contribution would be substantially strengthened and more broadly useful to both ML and power-systems audiences.

Cite this review as

---

## Round 0.3 · accepted · Accept

· Academic Editor

Accept

Dear authors thank you for addressing reviewers concerns.